JCB | Journal of Cell Biology

# NME3 binds to phosphatidic acid and mediates PLD6-induced mitochondrial tethering

You-An Su[1], Hsin-Yi Chiu[1], Yu-Chen Chang[1], Chieh-Ju Sung[1], Chih-Wei Chen[1], Reika Tei[2], Xuang-Rong Huang[1], Shao-Chun Hsu[3], Shan-Shan Lin[1], Hsien-Chu Wang[4], Yu-Chun Lin[4,5], Jui-Cheng Hsu[4], Hermann Bauer[6], Yuxi Feng[7], Jeremy M. Baskin[2], Zee-Fen Chang[1,8], and Ya-Wen Liu[1,8]

**Mitochondria are dynamic organelles regulated by fission and fusion processes. The fusion of membranes requires elaborative coordination of proteins and lipids and is particularly crucial for the function and quality control of mitochondria. Phosphatidic acid (PA) on the mitochondrial outer membrane generated by PLD6 facilitates the fusion of mitochondria. However, how PA promotes mitochondrial fusion remains unclear. Here, we show that a mitochondrial outer membrane protein, NME3, is required for PLD6-induced mitochondrial tethering or clustering. NME3 is enriched at the contact interface of two closely positioned mitochondria depending on PLD6, and NME3 binds directly to PA-exposed lipid packing defects via its N-terminal amphipathic helix. The PA binding function and hexamerization confer NME3 mitochondrial tethering activity. Importantly, nutrient starvation enhances the enrichment efficiency of NME3 at the mitochondrial contact interface, and the tethering ability of NME3 contributes to fusion efficiency. Together, our findings demonstrate NME3 as a tethering protein promoting selective fusion between PLD6-remodeled mitochondria for quality control.**

## Introduction

Mitochondria are highly dynamic organelles that continuously divide, traffic, and fuse. These processes regulate their size, shape, distribution, functional competence, and overall quality (Friedman and Nunnari, 2014; Mishra and Chan, 2016; Youle and van der Bliek, 2012). The fusion and fission of mitochondrial membranes are critical for their quality control and are orchestrated by specific proteins and lipids. Dynamin-related GTPase Drp1 catalyzes the fission of mitochondrial membranes for mitochondrial division, whereas Mfn1 and Mfn2 are responsible for outer membrane fusion and Opa1 for inner membrane fusion (Chan, 2006; Ramachandran, 2018). While these GTPases are the driving machinery for membrane remodeling, two non-bilayer-forming phospholipids, cardiolipin (CL) and phosphatidic acid (PA), have been found to be important for the fission and fusion processes of mitochondria (Choi et al., 2006; Huang et al., 2011; Kameoka et al., 2018; Osman et al., 2011).

CL is a negatively charged phospholipid comprising 10–15% of the total mitochondrial phospholipids and is synthesized in the mitochondrial inner membrane where it binds with and facilitates Opa1-mediated inner membrane fusion (Ban et al., 2017; Osman et al., 2011). A fraction of CL is further transported to the outer membrane upon inner membrane potential decrease and could bind with Drp1 to stimulate its mitochondrial fission activity or induce mitophagy via interacting with LC3 (Chu et al., 2013; Francy et al., 2017). Intriguingly, CL can be hydrolyzed and converted into PA by mitoPLD/PLD6, a phospholipase D on the outer membrane of mitochondria, where PA functions as a signaling lipid for mitochondrial clustering and facilitates fusion (Choi et al., 2006; Zhang et al., 2016). Although PA has been reported to inhibit mitochondrial fission by hindering the activity of Drp1 (Adachi et al., 2016), the direct downstream effectors of PA for mitochondrial tethering and fusion remain unclear.

Nucleoside diphosphate kinases, also known as the NME family, are a group of highly conserved proteins encoded by 10 genes, *NME1–NME10*, in humans (Desvignes et al., 2009). Among them, NME4 is located in the intermembrane space of mitochondria, where it exerts dual functions to charge Opa1 with GTP for inner membrane fusion or binds and externalizes CL to the outer membrane upon a mitophagic trigger (Chu et al., 2013; Kagan et al., 2016; Tokarska-Schlattner et al., 2008). On the other hand, NME3 is localized on the outer membrane of the

[1]Institute of Molecular Medicine, College of Medicine, National Taiwan University, Taipei, Taiwan; [2]Department of Chemistry and Chemical Biology and Weill Institute for Cell and Molecular Biology, Cornell University, Ithaca, NY, USA; [3]Imaging Core, College of Medicine, National Taiwan University, Taipei, Taiwan; [4]Institute of Molecular Medicine, National Tsing Hua University, Hsinchu, Taiwan; [5]Department of Medical Science, National Tsing Hua University, Hsinchu, Taiwan; [6]Department of Developmental Genetics, Max Planck Institute for Molecular Genetics, Berlin, Germany; [7]Department of Experimental Pharmacology, Medical Faculty Mannheim, Heidelberg University, Heidelberg, Germany; [8]Center of Precision Medicine, College of Medicine, National Taiwan University, Taipei, Taiwan.

Correspondence to Zee-Fen Chang: zfchang@ntu.edu.tw; Ya-Wen Liu: yawenliu@ntu.edu.tw.

mitochondria and plays a role in promoting mitochondrial fusion thus maintaining mitochondrial tubular morphology (Chen et al., 2019). In accordance, the loss of NME3 results in mitochondrial fragmentation and oxidative stress, leading to genome instability (Chen et al., 2020). Distinct from NME4, NME3 stimulates mitochondrial fusion in a kinase-independent manner, and the physiological importance of NME3 is highlighted by the association of homozygous mutation at the initiation codon of *NME3* in a newborn with fatal neurodegenerative disorder features (Chen et al., 2019).

NME3 lacks a transmembrane domain or a canonical mitochondrial targeting sequence. While investigating the mechanism of NME3 localization to the mitochondrial outer membrane, we found that the N-terminal region of NME3 binds to PA containing-liposome, and its enrichment at the mitochondrial contact interface depends on the activity of PLD6. We further showed that hexamerization is required for NME3 in tethering mitochondria in close apposition. Thus, NME3 fills the functional gap between PLD6-generated PA and the selective tethering of mitochondria that promote fusion.

## Results

### PLD6-induced mitochondrial clustering requires NME3 hexamerization on mitochondria

PLD6 facilitates mitochondrial fusion through its PA generation activity (Choi et al., 2006). Overexpression of PLD6 was sufficient to induce mitochondria clustering (Fig. 1, A and B), which depends on its catalytic activity (Fig. S1 A). Intriguingly, we found that NME3 depletion disrupted PLD6-induced mitochondria clustering (Fig. 1, A and B; and Fig. S1 B). Since an antibody for specifically endogenous NME3 is not available, we used CRISPR-knock-in in HeLa cells to tag the C-terminus of NME3 with an influenza hemagglutinin (HA) tag for detecting endogenous NME3 expression, which allowed us to test the siNME3 knockdown effect (Fig. S1 B). Similar results were also observed in mouse embryonic fibroblasts (MEFs) where the PLD6-induced mitochondria clustering is disrupted in NME3 knockout cells (Fig. S1, C and D). Of note, fragmented mitochondria that resulted from NME3 knockdown or knockout could both be rescued by NME3-GFP re-expression (Fig. S1, C–F).

Given that PLD6-induced mitochondrial tethering requires NME3, we then tested if NME3 binds to PA. To assess this possibility, we purified C-terminally His-tagged NME3 and performed the liposome flotation assay (Fig. 1 C). The results showed that NME3-His preferentially binds to liposomes composed of 20% PA but not to liposomes composed of 20% CL or 99% phosphatidylcholine (PC; Fig. 1, D and E). Together, these data suggest that NME3 binds to PA and functions as a downstream effector of PLD6.

We further compared the effect of the expression of GFP-tagged NME3 variants on mitochondrial morphology. Consistent with our previous findings, the results revealed that both wild-type and kinase-dead NME3[H135Q] could lead to mitochondrial hyper-clustering, whereas the E40D and E46D mutation that impairs NME3 hexamerization (Chen et al., 2019) had a rather modest effect on mitochondrial morphology (Fig. 1 F).

Contrarily, the expression of NME3[ΔN] defective in mitochondrial localization resulted in dispersed and shorter mitochondria (Fig. 1, F and G). The expression level of these NME3 mutants are shown in Fig. S1 G. These results suggest that NME3 can dictate mitochondrial morphology via its mitochondrial localizing and oligomerization activities.

### NME3 binds to PA and prefers lipid packing defects via its N-terminal region

We have previously shown that the N-terminal region of NME3 is responsible for its binding to the mitochondrial outer membrane. The amino acid alignment shows that the N-terminal 17-amino acid (N17) in NME3 is conserved in NME3 among vertebrates but not found in other members of the NME family (Fig. S2, A and B). The N17 region is composed of 12 hydrophobic amino acids, which give a hydrophobic moment with a measure of the amphipathicity of a helix, of 0.288 μH (calculated and derived from HeliQuest, Fig. 2 A). Expression of the GFP-tagged N17 localized to mitochondria without causing abnormal morphology, suggesting that the amphipathic helix of N17 is sufficient to confer NME3 mitochondrial membrane–binding activity (Fig. 2, A and B). Mutations of two phenylalanines, F9 and F13, of N17 to alanine to reduce the hydrophobic moment to 0.207 and 0.145 respectively, significantly reduced N17-GFP mitochondrial localization (Fig. 2, A and B).

Next, we wanted to characterize the N17 region in PA binding. N17 truncated or mutated NME3 proteins were purified for conducting PA-liposome flotation assay as described. The deletion of N17 or the mutation of F9/13A disrupted the binding between NME3 and PA liposome, while NME3[E40/46D] mutant protein retained PA binding activity (Fig. 2 C). Of note, the N17 truncation did not reduce the hexamerization ability of NME3[ΔN] as demonstrated by glutaraldehyde crosslinking analysis (Fig. S2 C). Most amphipathic helix-containing proteins display a curvature sensitivity, favoring binding to the highly curved membrane (Drin and Antonny, 2010; Giménez-Andrés et al., 2018). In agreement, NME3 also showed a preference for binding toward the curved membranes, i.e., liposomes with smaller diameters (Fig. 2, D and E). To examine the direct binding between N17 and PA, we purified N17-GFP and confirmed its preference toward PA liposome over other lipids, including PC, phosphatidylethanolamine (PE), phosphatidylserine (PS), and CL (Fig. S2 D).

The N17 region of NME3 lacks charged amino acids. We speculated that NME3 binds to PA preferentially through the lipid-packing defects resulting from cone-shaped lipids. To test this possibility, we prepared sonicated unilamellar vesicles (SUVs, with a smaller diameter than liposomes extruded from a membrane with 50 nm pore size, Fig. S2 E) composed of 100% PC to expose lipid packing defects for the NME3-liposome flotation assay. Data showed that NME3 also binds to PC liposomes with higher membrane curvature (Fig. 2, F and G). Taken together, these results demonstrate that NME3 binds directly to the membrane via its N17 amino acids with a preference for the membrane containing lipid-packing defects, such as the membrane enriched with cone-shaped phosphatidic acid or with high curvature.

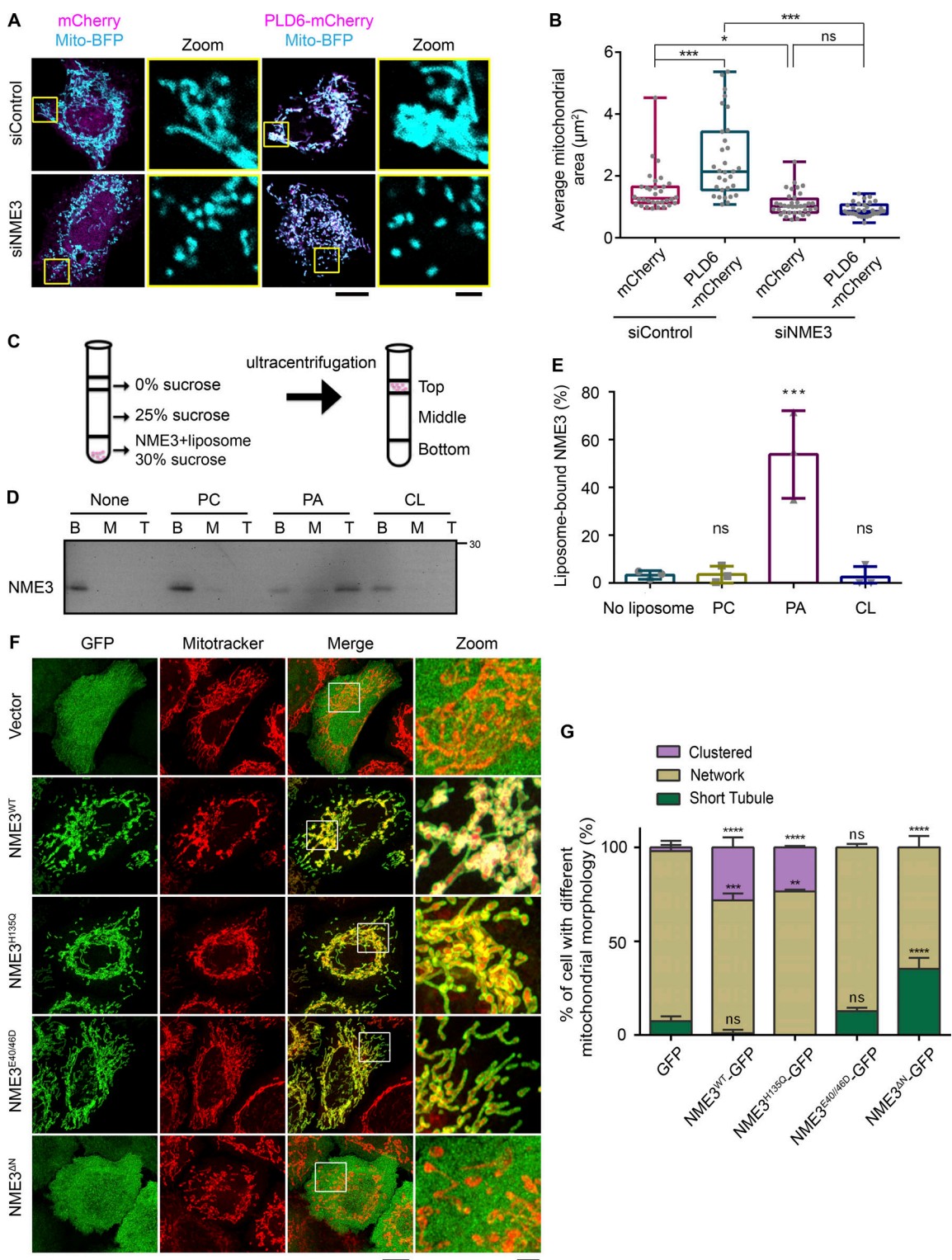

Figure 1. **NME3 binds to PA and is a downstream effector of PLD6. (A)** NME3 is required for the mitochondrial clustering effect induced by PLD6-mCherry overexpression. Control or NME3-depleted HeLa cells were transfected to overexpress mCherry or PLD6-mCherry. **(B)** The mitochondria in these cells were labeled by mito-BFP, and the average area of individual mitochondria was measured by ImageJ and shown in B. Each dot represents the averaged mitochondrial area of one cell. Scale bar, 10 μm. Bar for inset image, 2 μm. **(C)** Liposome flotation assays. NME3-His was incubated with or without liposome for 30 min. After binding, the suspension was adjusted to 30% w/v sucrose and overlaid with two layers of sucrose buffer and subjected to ultracentrifugation. After centrifugation, the top, middle, and bottom fractions were collected and analyzed by Western blotting. **(D)** NME3^WT binds to PA-containing liposomes. 1 μM NME3^WT was incubated with 300 μM, 100 nm liposome with different lipid compositions: 99% DOPC, 20% DOPA:79% DOPC or 20% CL:79% DOPC. **(E)** The proportion of NME3^WT in top fraction was quantified and compared in E. **(F)** Effects of exogenous NME3-GFP overexpression on mitochondrial morphology. HeLa cells transfected with indicated NME3-GFP constructs were labeled with a mitotracker and imaged with Airyscan confocal microscopy. Boxed areas were

magnified and shown aside. Scale bar, 10 μm. Bar for inset image, 2 μm. **(G)** Morphology of mitochondria were divided into three categories: short tubule, network, or clustered, to compare the effects of NME3-GFP on mitochondria from F. Data are expressed as mean ± SD of at least three independent experiments. Three different sets of experiments with more than 90 cells of each condition were quantified and analyzed with one-way ANOVA. *P < 0.05; **P < 0.01; ***P < 0.001; ns, not significant. Source data are available for this figure: SourceData F1.

### PLD6 facilitates the targeting of N17 of NME3 to mitochondria

Having shown the PA binding activity of the N17 region of NME3, we then asked whether the functional activation of PLD6 could affect the N17 polypeptide targeting to mitochondria. To this end, we utilized a recently developed, light-inducible optogenetic PLD system composed of a bacterial PLD that hydrolyzes PC to generate PA and is spatiotemporally regulated by a CRY2–CIBN light-mediated heterodimerization system (Tei and Baskin, 2020). By fusing the CIBN to a mitochondrial-targeting sequence, we could spatiotemporally increase PA on the mitochondrial outer membrane via inducing mitochondrial enrichment of optoPLD^WT with blue light conversion (Fig. S3 A). In agreement with previous reports of increased PA levels on mitochondrial membrane leading to mitochondrial clustering (Choi et al., 2006; Zhang et al., 2016), we found that mitochondria became more clustered after 70 min of photoactivation of

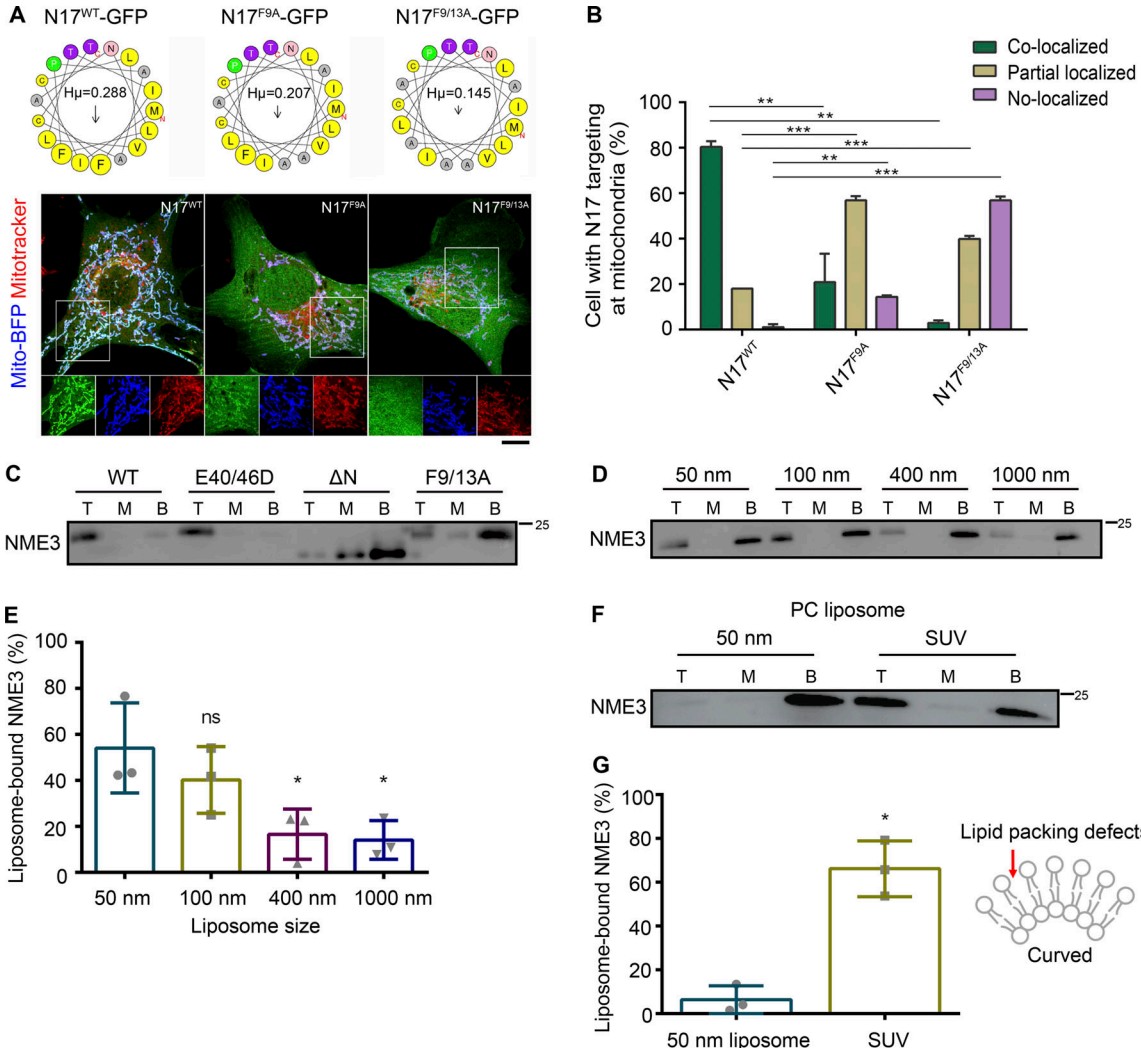

Figure 2. **The N17 region of NME3 is critical for mitochondrial and PA binding. (A)** N17 of NME3 is sufficient for mitochondria localization. Helical wheel projections of N17 and mutants are derived from HeliQuest (https://heliquest.ipmc.cnrs.fr/). **(B)** The GFP-tagged N17 and mutant constructs were co-transfected with mito-BFP, and their localization onto mitochondria are imaged, compared, and shown in B. Scale bar, 10 μm. **(C)** The N17 region of NME3 is essential for PA binding. 1 μM NME3-His with indicated mutations were incubated with PA liposome to examine their lipid binding ability. **(D)** NME3^WT prefers to bind liposomes with higher membrane curvature. 1 μM NME3^WT was incubated with 300 μM liposome composed of 10% DOPA:89% DOPC:1% rhodamine-PE but extruded with indicated membrane pore size. The larger the extrusion membrane pore size, the lower the liposome membrane curvature. **(E)** The proportion of NME3^WT in top fraction was quantified and compared. **(F)** NME3^WT binds to sonicated, small unilamellar vesicle (SUV) composed of 100% PC. **(G)** The proportion of NME3^WT in the top fraction was quantified and compared in G. Data are expressed as mean ± SD of three independent experiments and analyzed with one-way ANOVA. *P < 0.05; **P < 0.01; ***P < 0.001; ns, not significant. Source data are available for this figure: SourceData F2.

optoPLD$^{WT}$, whereas a catalytic-dead optoPLD$^{H170A}$ did not induce significant clustering (Fig. 3, A and B; and Fig. S3, B and C; and Video 1). In parallel, we observed a significant increase in N17-GFP targeting to mitochondria upon activation of optoPLD$^{WT}$, but not the catalytic dead optoPLD$^{H170A}$ (Fig. 3, C–E).

Next, we tested the effect of PLD6 depletion on N17-GFP binding to mitochondria by microscopy analysis (Fig. 3, F and G; and Fig. S3 D) and mitochondrial fractionation (Fig. S3 E). Similar to a previous report, PLD6 depletion resulted in mitochondrial fragmentation that could be restored by PLD6-mCherry re-

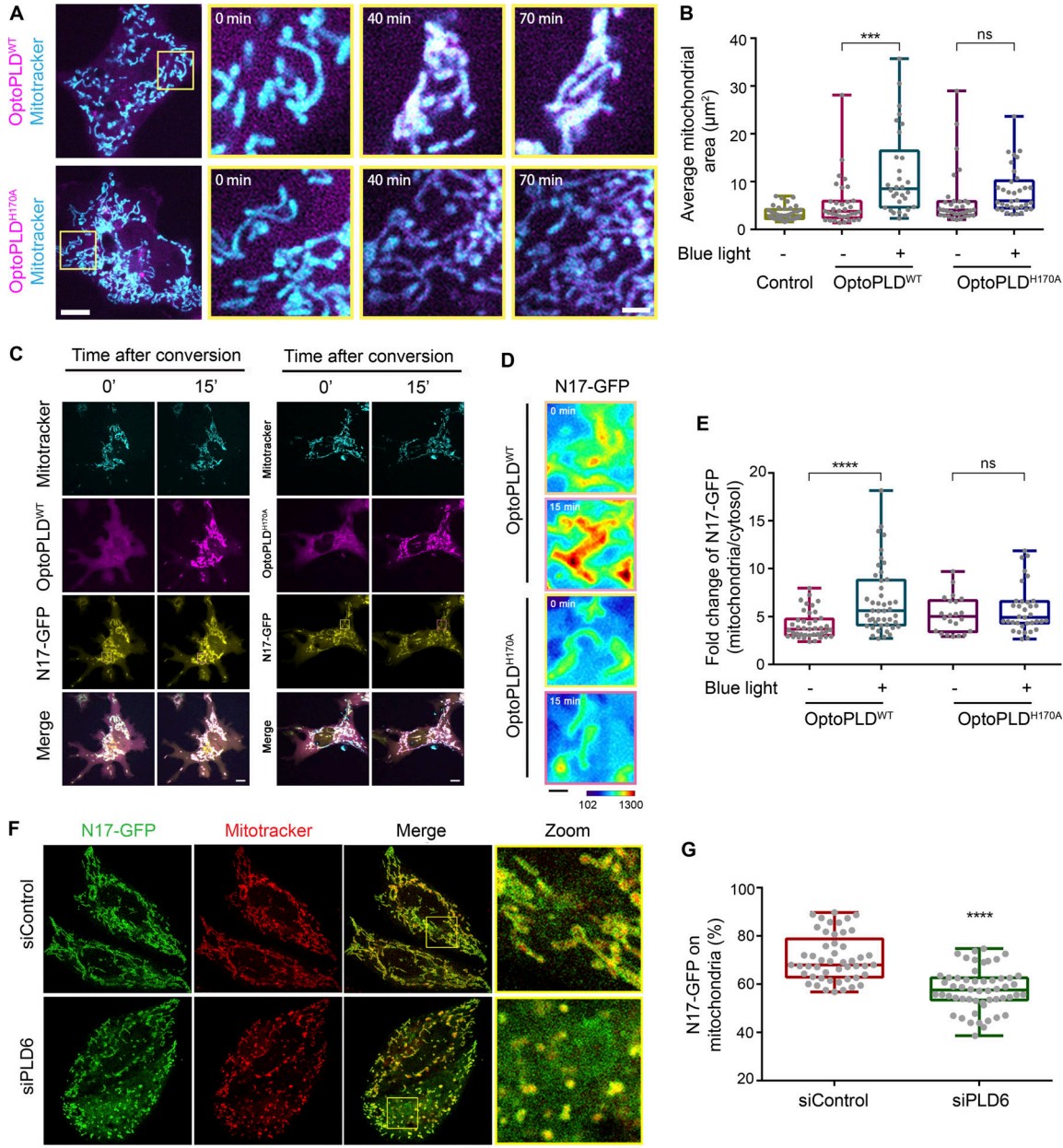

Figure 3. **Mitochondrial PA is required for the subcellular location of N17-GFP. (A)** The increase of mitochondrial PA by targeting optoPLD$^{WT}$ onto mitochondria induces their clustering. optoPLD$^{WT}$ or optoPLD$^{H170A}$ were transfected into Cos-7 cells, and these cells were stained with mitotracker deep red prior to conversion. After 70 min of blue-light-induced mitochondrial targeting by pulse with 90 s intervals, the effect of optoPLD$^{WT}$ on mitochondrial clustering was examined and quantified by the signal of mitotracker deep red. Scale bar, 10 µm. **(B)** Bar for inset image, 2 µm. The average mitochondrial area was quantified, compared to control cells, and shown in B. Each dot represents the average mitochondrial area of one cell. **(C)** optoPLD$^{WT}$ enhances the mitochondrial localization of N17-GFP. After the co-transfection of N17-GFP and optoPLD$^{WT}$ into Cos-7, the mitochondria were labeled with mitotracker deep red and subjected to blue-light conversion for 45 min. **(D and E)** Scale bar, 10 µm. The intensity of N17-GFP on mitochondria was monitored, magnified and shown (D), quantified and compared with optoPLD$^{H170A}$ by ImageJ (E). Bar for heatmaps, 2 µm. To quantify the mitochondrial targeting efficiency, the total intensity of N17-GFP on mitochondria in a cell was calculated by ImageJ and divided by the total N17-GFP intensity of a cell. Each dot represents the ratio of one cell. **(F)** PLD6 depletion reduces the mitochondrial localization of N17-GFP. The proportion of N17-GFP co-localized with mitotracker was measured in HeLa cells depleted with PLD6. Scale bar, 10 µm. Bar for inset images, 2 µm. **(G)** The mitochondrial localization efficiency of N17-GFP was quantified. Data are expressed as mean ± SD of at least three independent experiments and analyzed with one-way ANOVA. *P < 0.05; **P < 0.01; ***P < 0.001; ns, not significant.

expression (Fig. S3, F and G). Furthermore, PLD6 depletion reduced N17-GFP binding to mitochondria, but the extent of reduction was rather moderate. Two possibilities could contribute to the moderate reduction. One is that the amount of PA on the mitochondrial outer membrane was not generated mainly by PLD6. Another possibility is that other factors in mitochondria can also adopt N17-GFP binding. To address that, we tested whether full-length of NME3 localized on mitochondria is affected by PLD6-generated PA. It turned out that the overall amounts of endogenously or exogenously expressed NME3 on mitochondria were not significantly decreased by PLD6 depletion (Fig. S4, A–H). We then replaced the N17 region of NME3 with an amphipathic helix derived from the PA-binding domain of Spo20 (Horchani et al., 2014). Surprisingly, this replacement disrupted the mitochondrial targeting ability of NME3 (Fig. S4 I). Apparently, the mitochondria binding activity of NME3 is not solely determined by PA binding but requires the interaction with other factor(s) on the mitochondrial outer membrane. NME3 plays a critical role in mitochondrial dynamics, and NME3 depletion causes mitochondrial fragmentation due to the loss of fusion (Chen et al., 2019). Considering PLD overexpression that increases PA can enhance N17-GFP binding, the above findings evoke a question of whether the PA binding function mainly mediates NME3 enriched at the sites specified for the fusion event.

### PLD6 causes the enrichment of NME3-GFP at the mitochondrial contact interface

We then focused on when or where NME3 binding to mitochondria is affected by PLD6 function. PLD6 has been proposed to act as a dimer that catalyzes PA production in trans, i.e., on the opposing mitochondria (Choi et al., 2006; Osman et al., 2011). Accordingly, the effect of PLD6 depletion on NME3 binding at the mitochondrial contact interface was examined. Herein, we used time-lapse microscopy to monitor the spatial distribution of NME3-GFP on mitochondria. An acute enrichment of NME3-GFP at two closely opposed mitochondria was observed (Fig. 4 A and Fig. S4 J, arrows, and Video 2). The fluorescent intensity analysis revealed that the amount of NME3-GFP at the mitochondrial contact interface was 1.79 ± 0.35 fold higher than the average intensity in the mid-zone of mitochondria (Fig. 4, B and C). More importantly, the enrichment of NME3-GFP at mito-mito contact interface was significantly reduced by PLD6 depletion from 1.7 ± 0.39 down to 1.3 ± 0.29 fold (Fig. 4, E–G), demonstrating that PLD6 affects NME3 enriched at the contact interface. Of note, the PA binding mutant NME3$^{F9/13A}$, as well as the hexameric mutant NME3$^{E40/46D}$, showed significantly lower enrichment at the mitochondrial contact interface (1.43 ± 0.26 and 1.49 ± 0.24, respectively; Fig. 4, B and C).

We further quantified the fusion event after 1 min of mitochondrial contact and found that mitochondria with NME3-GFP expression have higher probability (29.3%) for fusion than cells expressing TOM20-GFP (15.8%), NME3$^{E40/46D}$-GFP (9.5%), or NME3$^{F9/13A}$-GFP (3.0%; Fig. 4 D). Here, it should be emphasized that NME3$^{F9/13A}$-GFP lacks PA binding activity but is still capable of mitochondrial localization. Altogether, these data suggest that PLD6-generated PA drives a spatial

control of NME3 enrichment at mitochondrial contact sites to facilitate membrane fusion.

### The role of hexamerization in NME3-mediated mitochondrial membrane tethering

NME3$^{E40/46D}$-GFP expression retains PA binding function but still affects the fusion event (Fig. 2 C and Fig. 4 D). Since E40/46D mutation affects the hexameric status of NME3, it is likely that hexamerization and PA binding functions cooperate to drive mitochondrial membrane tethering. We further expressed Neon-FKBP-NME3$^{ΔN}$ with Tom20-CFP-FRB. Upon rapamycin addition, Neon-FKBP-NME3$^{ΔN}$ was targeted onto the mitochondrial outer membrane (Fig. 5, A and B; Komatsu et al., 2010). After 13.5 min, cells expressing Neon-FKBP-NME3$^{ΔN-WT}$ and $^{-H135Q}$ exhibited similarly clustered and aggregated mitochondrial networks (Fig. 5 C and Video 3). As a comparison, rapamycin-induced mitochondrial translocation of NME3$^{ΔN-E40/46D}$ did not cause such fast and prominent mitochondrial clustering and aggregation. Given that E40 and E46 positioned at the interface of two NME3 trimers are important residues for hexamer formation, these results demonstrate that molecular hexamerization of NME3 on the mitochondrial outer membrane promotes mitochondrial membrane tethering.

We further performed liposome tethering assays to substantiate this notion. After incubation with purified NME3-His, the distribution of liposomes composed of 20% PA, 79% PC, and 1% rhodamine-PE was observed under fluorescent microscopy. Interestingly, NME3$^{WT}$ resulted in a dose-dependent liposome clustering phenotype, but not the NME3$^{E40/46D}$ (Fig. 5 D). Dynamic light scattering (DLS) analysis was used to measure the size distribution of 100-nm liposomes containing 20% PA. The size of liposomes was significantly increased by incubation with NME3$^{WT}$, but not hexameric-defective mutant NME3$^{E40/46D}$ nor the lipid binding defect mutant NME3$^{ΔN}$ (Fig. 5, E and F). Negative stain transmission electron microscopy further confirmed that NME3 tethers liposomes into a cluster instead of driving fusion (Fig. 5, G and H).

Furthermore, we performed in vitro mitochondrial tethering assay to verify the role of NME3 in mitochondrial membrane tethering via hexamerization. To avoid the complication of Mfn1 and Mfn2 in the mitochondrial tethering and fusion process, Mfn1 and 2 were simultaneously depleted in cells transfected with the plasmid of mito-mCherry or mito-YFP in combination with the expression vector of NME3$^{WT}$ or NME3$^{E40/46D}$. The mCherry- and YFP-labeled mitochondria were separately isolated from these cells and mixed for in vitro tethering assay as illustrated in Fig. S5 A. Stable tethered green/red mitochondria were evaluated by time-lapse confocal microscopy (Fig. 5 I). The quantitation data showed that the presence of NME3$^{WT}$ significantly promotes stable tethering of separate mitochondria, while NME3$^{E40/46D}$ mutant did not give a tethering effect (Fig. 5 J and Video 4).

A recent report has shown that hypotonic cell swelling can reveal inter-organelle contacts and protein concentrated at the contact sites (King et al., 2020). Using this approach, Tom20-GFP-expressing mitochondria that display modest clustering of large intracellular vesicles (LICVs) were observed after 10 min of

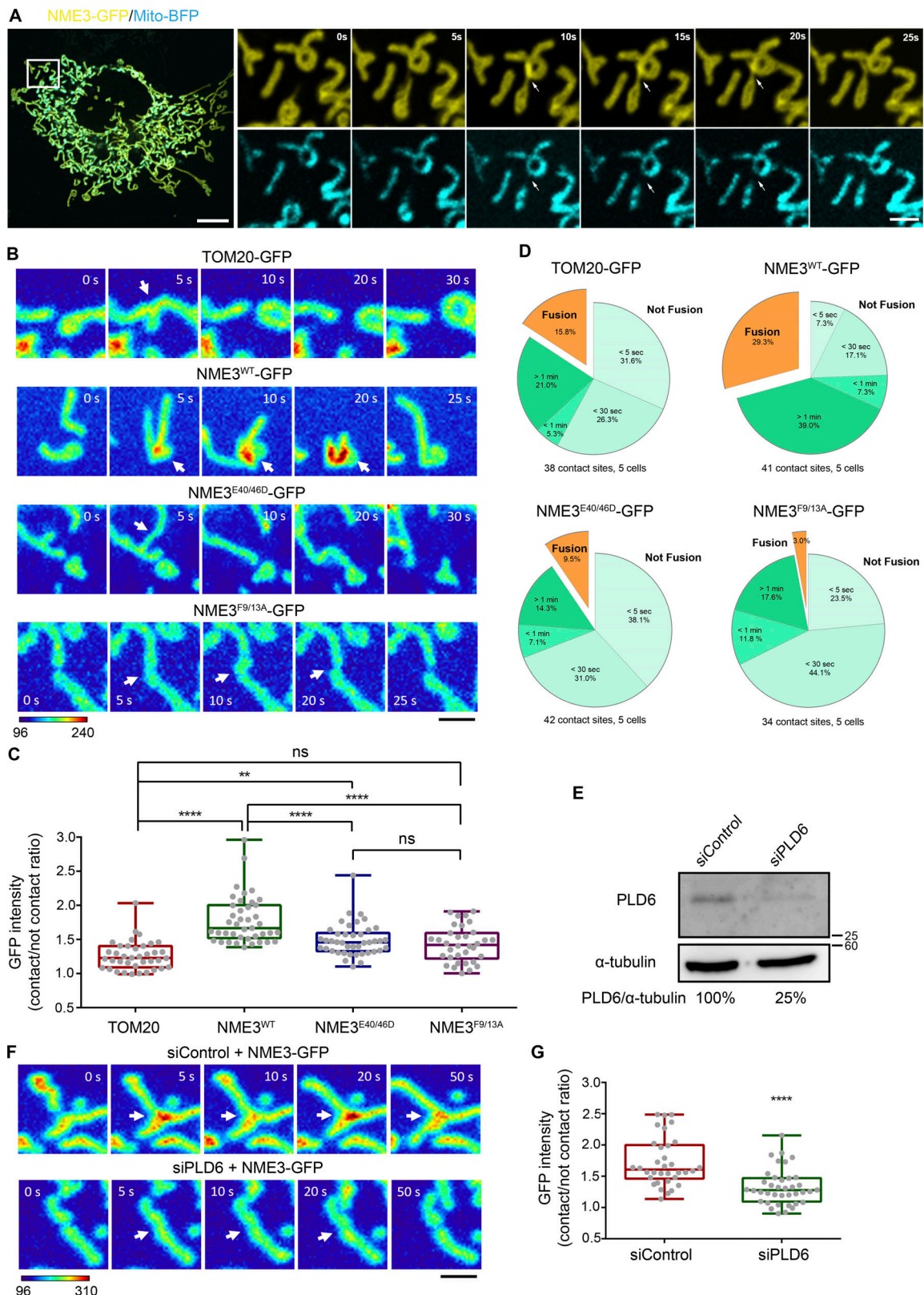

Figure 4. **NME3 is enriched at the mitochondrial contact interface dependent on PLD6. (A)** The enrichment of NME3-GFP at mitochondrial contact interface prior to fusion. The dynamic distribution of NME3-GFP in two contact mitochondria was observed in Cos-7 cells co-transfected with NME3-GFP and mito-BFP which labels the mitochondrial matrix. Arrows indicate the contact interface between two closely positioned mitochondria. Scale bar, 10 μm. Bar for inset images, 2 μm. **(B)** The enrichment of different mutant NME3-GFP at the mitochondrial contact interface. The GFP intensity of these constructs was converted to a heat map. Scale bar, 2 μm. **(C)** The intensity of GFP-tagged proteins at the mitochondrial contact interface was quantified, normalized with the non-contact area, compared to the control mitochondrial outer membrane protein, TOM20, and shown in C. **(D)** The following fate of two contact

mitochondria after 1 min was analyzed and shown in D. **(E)** Knock-down efficiency of PLD6. HeLa cells were transfected with scrambled or PLD6 siRNA SMARTPool, and the knockdown efficiency was quantified by Western blotting and shown below the panel. **(F)** PLD6 depletion perturbs the enrichment of NME3-GFP at the mitochondrial contact interface. **(G)** NME3-GFP expressed in control or PLD6-depleted HeLa cells was imaged by time-lapse microscopy, quantified as described in B and shown in G. Scale bar, 2 μm. Data are expressed as mean ± SD of three independent experiments and analyzed with one-way ANOVA. *P < 0.05; **P < 0.01; ***P < 0.001; ns, not significant. Source data are available for this figure: SourceData F4.

hypotonic buffer incubation. Under the same condition, we found that mitochondria in cells expressing NME3$^{WT}$ and NME3$^{H135Q}$-GFP display abundant clustered LICVs in cells, but not in NME3$^{E40/46D}$-, NME3$^{F9/13A}$-, or NME3$^{\Delta N}$-expressing cells (Fig. S5 B). Importantly, both NME3$^{WT}$ and NME3$^{H135Q}$-GFP display clear enrichment at the contact interface between swelled mitochondria (Fig. S5 C). Furthermore, the mito-mito contact angle and length, a measure of surface tension and affinity respectively, are also higher between mitochondria expressing NME3$^{WT}$ and NME3$^{H135Q}$-GFP than Tom20-GFP (Fig. S5, D and E). Together, these results suggest a model of NME3 presenting on the mitochondrial outer membrane, i.e., NME3 in hexameric form has N-terminal regions facing outward for mitochondrial outer membrane binding. One side of the hexamer stably associates with the mitochondrial outer membrane, while the other side is readily available for recognizing PA on the opposing mitochondrial outer membrane. Thereby, these properties make NME3 capable of tethering the mitochondrial membrane with specific lipid-packing defects.

### Nutrient starvation promotes the enrichment of NME3 at the contact interface and mitochondrial fusion

Mitochondrial morphology and activity are tightly regulated to meet different cell stresses or demands. Nutrition depletion has been reported to result in mitochondrial elongation that equips mitochondria for maximal energy production and protects mitochondria from autophagosomal degradation (Rambold et al., 2011). Furthermore, we previously found that NME3-deficient patient fibroblasts lost glucose starvation–induced mitochondrial elongation and died after 32 h of starvation (Chen et al., 2019). We thus wonder if the enrichment of NME3 would be improved by nutrient starvation. Therefore, we measured the NME3-GFP intensity in the mitochondrial contact interface and found it was significantly increased from 1.48 ± 0.48 to 1.85 ± 0.76 fold change under nutrient starvation (3 h) than in the control medium (Fig. 6, A–C). Furthermore, the fusion probability upon mitochondrial contact was also higher in cells cultured in the nutrient-starved medium than in the control medium, 39.2% and 28.6%, respectively (Fig. 6 D). Thus, short-term nutritional depletion increases NME3-GFP enrichment at the mitochondrial contact interface, which could promote the subsequent fusion efficiency of mitochondria.

## Discussion

Tethering proteins play essential roles in organelle functions and vesicular transport, including stabilization of organelle membrane contact sites, docking of vesicles for accurate trafficking, as well as guiding SNARE proteins assembly for efficient fusion (Hong and Lev, 2014; Song and Wickner, 2019). Here, we

discover that NME3 is a mitochondrial tethering protein that binds to the outer membrane remodeled by PLD6. We proposed that the N-terminal sequence of NME3 is an amphipathic lipid packing sensor (ALPS) motif that enables NME3 enrichment at the mitochondrial contact interface where PLD6 generates PA on the opposing mitochondrial membrane (Fig. 6 E). After binding to PLD6-generated PA on mitochondrial membranes at close apposition, the lateral interaction of hexameric NME3 on contact sites stabilizes mitochondrial membrane association. It is well documented that mitofusins by themselves have a tethering function via HR2-mediated dimerization (Koshiba et al., 2004). Therefore, mitofusins can tether and then fuse the mitochondrial outer membrane. Since mitofusins have mitochondrial transmembrane domains, it is unclear how the mitofusins-mediated tethering process achieves the selectivity for unhealthy mitochondria. We have previously shown that NME3 can also interact with Mfns (Chen et al., 2019). Herein, we depicted a model in which the N-terminal motif of NME3 hexamer selects the PA signal derived from CL externalized to the outer mitochondrial membrane for tethering, which bolsters Mfns-mediated tethering and fusion of unhealthy mitochondria (Fig. 6 E).

PA is a fusogenic lipid and is assumed to promote mitochondrial fusion by its unique geometry and its ability to inhibit Drp1 (Adachi et al., 2016; Choi et al., 2006). On the other hand, CL could stimulate Drp1 activity and drive mitochondrial division, mitophagy, or apoptosis when it is externalized onto the mitochondrial outer membrane (Pizzuto and Pelegrin, 2020; Schlattner et al., 2018). Therefore, CL on the outer membrane could function as an "eat-me" signal for Drp1 and mitophagic machinery (Chu et al., 2013). Inversely, PLD6-generated PA might thus serve as a "fuse-me" signal, which is recognized by NME3, to proceed with membrane tethering. Given that CL externalization is triggered by the decrease of mitochondrial membrane potential, which could be induced by nutrient starvation or mitochondrial damage, we speculate that PLD6 and NME3 may be responsible for the tethering and fusion of mitochondria with lower membrane potential and externalized CL. Furthermore, these findings raise questions about what is the molecular decision for CL conversion to PA by PLD6 and whether there is a threshold of CL as the eat-me signal.

It should be mentioned that PA is present on the membrane in different organelles. So, why is the majority of NME3 and N17-GFP localized on the mitochondrial membrane? Evidently, the amphipathic helix of N-terminal 17 amino acids confers additional properties to mediate the binding to the mitochondrial outer membrane. In this study, the N-terminal region of NME3 replaced with ALPS from other proteins disables NME3 localized on mitochondria, suggesting that the N17 region mediates the binding of NME3 mainly through other factor(s) on

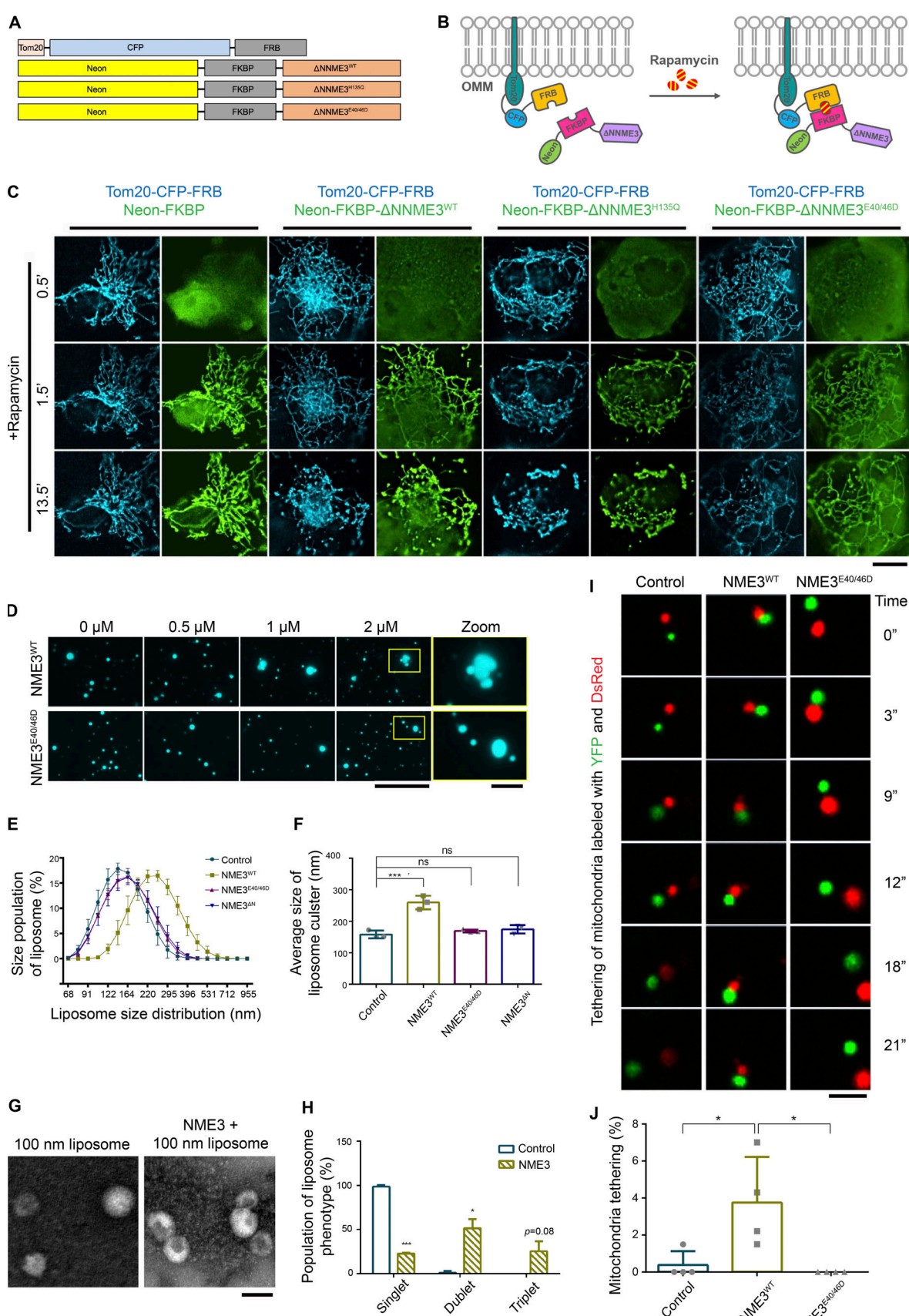

Figure 5. **NME3 tethers liposomes or mitochondria into cluster. (A)** NME3 constructs used in this rapamycin-induced mitochondrial targeting of NME3$^{\Delta N}$. **(B)** Scheme of the rapamycin-induced mitochondrial targeting assays. **(C)** Cos-7 cells co-transfected with Tom20-CFP-FRB and indicated Neon-FKBP-ΔNNME3

constructs were imaged upon rapamycin addition. Scale bar, 10 μm. **(D)** Liposome tethering assay. Purified NME3$^{WT}$ or NME3$^{E40/46D}$ were incubated with 10 μM liposome (20% DOPA:79% DOPC:1% rhodamine-PE) extruded from 1,000 nm pore size. After 30 min incubation, the distribution of liposome was imaged with fluorescent microscopy. Boxed areas were magnified and shown. Scale bar, 10 μm. Bar for inset images, 2 μm. **(E)** Dynamic light scattering assay. 20 μM, 100 nm liposomes were incubated with 2 μM indicated NME3 and analyzed with dynamic light scattering. **(F)** The distribution of liposome size was shown in E, and the average size of liposome were displayed in F. **(G)** Negative stain TEM of NME3$^{WT}$-liposome cluster. **(H)** The population of liposome tethers was quantified. Bar, 200 nm. **(I)** Mitochondria tethering assay. The representative time-series images for mitochondria expressing different NME3-HA constructs. The DsRed-labeled mitochondria bound with YFP-labeled mitochondria for more than 15 s were considered stably tethered mitochondria. **(J)** The percentage of tethered mitochondria were quantified and shown in J. More than 200 mitochondria were count ($n$ = 4). Scale bar, 5 μm. Data are expressed as mean ± SD of at least three independent experiments. *$P$ < 0.05; ***$P$ < 0.001; ns, not significant.

the outer membrane rather than PA binding. Given that NME3 is capable of interacting with other mitochondrial proteins such as Mfn1/2 (Chen et al., 2019), it is possible that the maintenance of NME3 on the mitochondrial membrane is through interaction with other mitochondrial proteins containing transmembrane domain. Based on our observation, PLD6-mediated PA generation at the mitochondrial interface gives a spatial signal for NME3 enrichment that promotes membrane tethering for the subsequent fusion process.

Several members of the NDPK family have been identified to function as GTP fueling enzymes via its kinase activity for many dynamin-related GTPases in several membrane remodeling processes, including endocytosis, mitochondrial inner membrane fusion, and peroxisomal division (Boissan et al., 2014; Imoto et al., 2018). Interestingly, although N17 of NME3 is conserved in vertebrates, the NME3 homolog in a red alga, DYNAMO1, is not equipped with an amphipathic helix at its N-terminus (Imoto et al., 2018). Since DYNAMO1 exerts functions on the division of mitochondria and peroxisomes, whether the N-terminal amphipathic helix of vertebrate NME3 is evolved to sense lipid signals for mitochondrial quality control remains to be investigated.

NME3 deficiency has been shown to associate with neuronal degeneration disorder. The cellular level of NME3 is very low, but it has been detected in mitochondria-derived vesicles and the interactome of Sars-CoV2 proteins (König et al., 2021; Liu et al., 2021). These findings warrant the importance of understanding the cryptic function and the membrane binding mechanism of NME3 in the maintenance of health.

## Materials and methods
### Cell culture, transfection, and lentiviral infection
Cos-7 and HeLa cells were cultured with DMEM, 10% FBS, and antibiotics. For transfection, cells at 70% confluency were transfected with interested DNA using Lipofectamine 3000 (Invitrogen) or TransIT (Mirus Bio), as recommended by the manufacturer. For lentiviral infection, 50% confluent myoblast were infected with viruses together with 8 μg/ml polybrene and selected with 2 μg/ml puromycin for 4 d followed by transfection and microscopy analysis.

To generate endogenous NME3 C-terminal labeling with fluorescent EGFP tag, HeLa cells were transfected with donor plasmid (pUC19-hNME3-EGFP-KI), Cas9-mcherry that contains sgRNA fusion plasmid by Turbofect transfection reagent. After 48 h of transfection, single-cell sorting for cells expressing fluorescent marker EGFP was conducted. We then utilized

confocal microscopy, Western blot, PCR, and sequencing to validate the NME3-EGFP knock-in cells.

### Molecular biology
For NME3 and PLD6 expression, human NME3 and PLD6 were cloned into pEGFP-N1 or pmCherry-N1, respectively, for expression in mammalian cells. Different mutants were generated with site-direct mutagenesis. Plasmids used in this study are listed in Table S1. Lentiviral shRNA was generated and performed as described (Chuang et al., 2019), and the targeted sequences are listed in Table S2.

For mitochondria-targeted optoPLD$^{WT}$ construction, we utilized the plasma membrane-optoPLD$^{WT}$ constructs (#140061; Addgene) developed previously (Tei and Baskin, 2020) but with the organelle-targeting sequence replaced with a mitochondrial targeting sequence (MTS), the transmembrane domain from OMP25 (RGDGEPSGVPVAVVLLPVFALTLVAVWAFVRYRKQL), at the C terminus of CIBN, to obtain bicistronic vectors expressing CRY2-mCherry-PLD$_{PMF}$-P2A-CIBN-MTS or the catalytic-dead mutant.

### Fluorescent microscopy
For mitochondrial labeling, cells were incubated with a medium containing 150 ng/ml Mitotracker Red CMXRos (Invitrogen) for 30 min at 37°C. After washing, cells were fixed with 4% formaldehyde. Samples were observed under a confocal microscope LSM700 with 63×, 1.35-NA oil-immersion objective (Carl Zeiss).

For the rapamycin-inducible mitochondrial targeting experiment, Cos-7 cells were treated with 100 nM rapamycin (LC laboratories) and imaged under Nikon T1 inverted fluorescence microscope (Nikon). For optoPLD$^{WT}$ conversion, Cos-7 cells were imaged under spinning disc confocal microscope with 63×, 1.40-NA oil-immersion objective (Carl Zeiss) and Photometrics Prime 95B sCMOS camera in an imaging medium (phenol-red free DMEM with 10 mM HEPES pH 7.4 and 10% FBS) and converted by light with 488 nm wavelength, 100% laser, 150 ms exposure time for 70 min with a 90-s time interval.

For live cell imaging, Cos-7 and HeLa cells were imaged under a spinning disc confocal microscope (Carl Zeiss) in an imaging medium. Time-lapse imaging was performed 5 s per frame for 10–30 min with a single focal plane and autofocus. To image NME3-GFP distribution in cells under starvation, cells were starved in a nutrient depletion medium (DMEM without FBS and $L$-glutamine and supplemented with 10 mM HEPES). Cells were imaged with a spinning disc confocal microscope (Carl Zeiss) after 3 h of culture in a nutrient-starvation medium with 5 s per frame for 10–15 min.

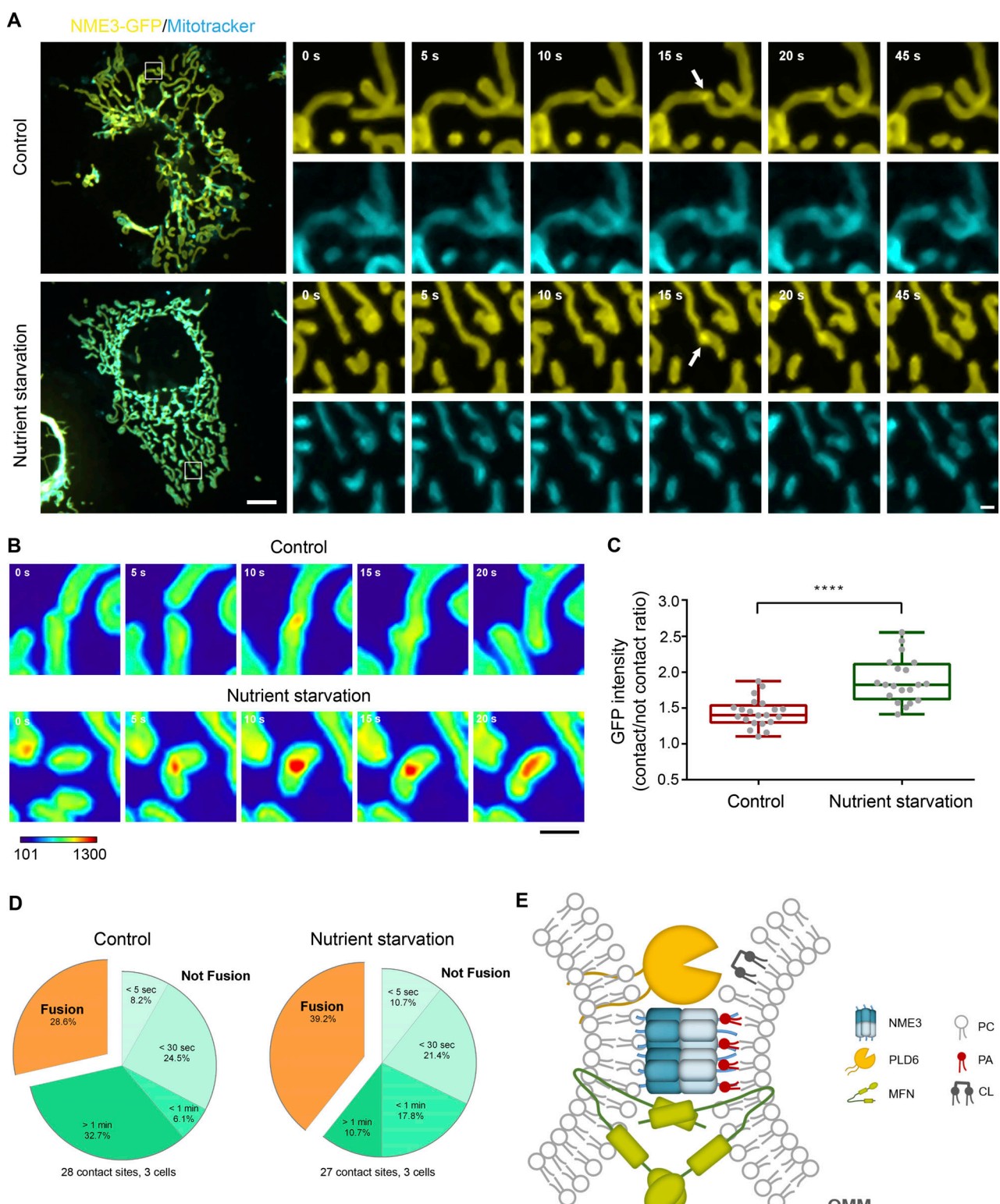

Figure 6.   **Nutrient starvation enhances NME3-GFP enrichment at the mitochondrial contact interface. (A)** The enrichment of NME3-GFP at mitochondrial contact interface upon nutrient starvation. The dynamic distribution of NME3-GFP was observed in Cos-7 cells cultured in control or nutrient-starved media. Arrows indicate the contact interface between two closely positioned mitochondria. Scale bar, 10 μm. Bars for inset images, 2 μm. **(B)** Effect of nutrient starvation on the enrichment of NME3-GFP at the mitochondrial contact interface. The GFP intensity of these constructs was converted to a heat map. **(C)** The intensity of GFP-tagged proteins at the mitochondrial contact interface was quantified, normalized with the non-contact area, compared to the intensity enrichment of cells cultured in control medium. Scale bar, 2 μm. **(D)** Data are expressed as mean ± SD of three independent experiments analyzed with Student's *t* test. ****P < 0.0001. The following event of two contact mitochondria was analyzed. **(E)** A hypothetical mechanism for PLD6 and NME3-mediated tethering of CL-externalized mitochondria. When two mitochondria with CL externalized onto its outer membrane encounter, PLD6 could catalyze CL into PA

in trans at the mitochondrial contact interface. The PA then enriches NME3 via intercalating the amphipathic helix of N-terminal 17 amino acids of NME3 into the lipid packing defects. Through its oligomeric structure and membrane binding ability, NME3 functions as a selective tethering factor to stabilize mitochondrial apposition thus facilitate fusion of damaged mitochondria.

## Imaging quantification

We utilize Matlab to quantify the mitochondrial targeting efficiency of our interested proteins. Segmentation of cells was done automatically in Matlab using edge detection followed by dilation and hole-filling. Postprocessing was performed to remove small objects and connected objects on the border. Segmentation of mitochondria was done by Mitometer (Lefebvre et al., 2021). To quantify the mitochondrial targeting efficiency, the total intensity of N17-GFP or NME3-GFP on mitochondria was calculated and divided by the total N17-GFP or NME3-GFP intensity inside cells.

To quantify the enrichment of protein at the interface of two closely opposed mitochondria, the NME3-GFP intensity was measured by line scan analysis in ZEN software (Carl Zeiss). One contact site intensity divided by the average of five mitochondria midzone intensity in either of two mitochondria is the GFP intensity fold change. GFP intensity was normalized to the background.

## Protein purification and liposome preparation

NME3-His was expressed from BL21 after 3 h of 0.1 mM IPTG induction at 30°C. Proteins were purified with HisPur Ni-NTA Resin beads (Thermo Fisher Scientific) dialyzed into storage buffer (20 mM HEPES pH 7.5, 150 mM KCl, 1 mM Dithiothreitol). Dialyzed proteins were centrifuged at $20,640 \times g$ for 30 min before storage at –80°C.

For liposome preparation, indicated lipid mixtures were dried, rehydrated in buffer containing 20 mM HEPES pH 7.5, 150 mM KCl, and subjected to a series of freeze–thaw cycles before extrusion through polycarbonate membranes (Whatman) with a pore size ranging from 50 to 1,000 nm using Avanti Mini-Extruder.

## Liposome flotation and tethering assays

A liposome flotation assay was conducted as previously described (Bigay et al., 2005). Briefly, 1 µM purified NME3-His was incubated with 300 µM liposomes for 20 min at 37°C. Mixtures were added with 75% sucrose (in 20 mM HEPES pH 7.5, 150 mM KCL buffer) for the final 500 µl 30% sucrose concentration (bottom fraction). All mixtures were placed at the bottom of a thin wall polypropylene tube (Beckman), followed by 800 µl 25% sucrose addition (middle fraction) and 100 µl, 20 mM HEPES pH 7.5, 150 mM KCL buffer (top fraction). Middle and top fractions were added carefully to prevent disturbing fraction boundaries. Samples were centrifuged with precooled SW 55 Ti Swinging-Bucket Rotor (Beckman) at $240,000 \times g$, OPTIMA L-90K Ultracentrifuge (Beckman Coulter) at 4°C for 2 h. After centrifugation, indicated samples were collected and analyzed by Western blotting with anti-His antibodies.

For liposome tethering analysis, a µ-slide eight-well chamber (iBidi) was coated with 1% BSA for 30 min. 10 µM, 20% PA, and 1,000 nm-liposomes were incubated with 0, 0.5, 1, or 2 µM purified NME3 at 37°C for 30 min. Liposomes–protein mixture was transferred to precoated chamber and observed with an inverted fluorescent microscope (Axio Observer Z1, Zeiss). For DLS analysis, 20 µM, 100 nm liposomes containing 20% PA were incubated with 2 µM NME3 for 30 min at 37°C. The mixture was transferred to a disposable Polystyrene (PS) cuvette (ratiolab) and analyzed by MALVERN Nano-ZS.

## Transmission electron microscopy

25 µM, 20% PA, and 100 nm liposomes were incubated with or without 5 µM NME3$^{WT}$ for 30 min at room temperature. The mixture was then adsorbed onto carbon-coated, glow-discharged grids (200 mesh copper) and stained with 2% uranyl acetate. Images were captured by Hitachi H-7650 EM at 75 kV and a nominal magnification of 120,000×.

## Mitochondrial fractionation and mitochondrial tethering assay

The procedure of mitochondrial fractionation and mitochondrial tethering analysis is based on a published paper with few modifications (Ban et al., 2017). HEK293T cells were transfected with siRNA against MFN1/2 (M-010670-01 for siMFN1, M-012960-00 for siMFN2; Dharmacon) for 2 d. An equal amount of the expression plasmids of NME3-HA (empty vector control, NME3$^{WT}$, or NME3$^{E40D/E46D}$) and mito-YFP or mito-DsRed were transfected to the 293T with inhibited MFN1/2 for 15 h. For mitochondrial fraction, cells were scraped from dishes with 0.5 ml of prechilled homogenization buffer (HB) containing HEPES-KOH (10 mM, pH 7.4), 220 mM mannitol, 70 mM sucrose, and 1% protease inhibitors. Cells were homogenized with 40 strokes in a 1-ml tissue homogenizer with a loose pestle. The unbroken cells and cellular debris were removed by centrifugation at $800 \times g$ for 5 min at 4°C. The mitochondria-enriched pellets were pelleted by centrifugation at $10,000 \times g$ for 10 min at 4°C and then washed by pre-chilled HB twice. Resuspended in 100 µl of HB, the mitochondrial-enriched pellets were broken into smaller mitochondria by sonication using a bioruptor (UCD-200; Diagnode, 10 s at low intensity). After removing the larger pellet by centrifugation at $800 \times g$ for 5 min at 4°C, resuspended mitochondria labeled with mito-YFP or mito-DsRed were mixed with equal volume and then incubated for 1 h at 30°C for mitochondria binding. The binding of mitochondria was detected by using confocal fluorescence microscopy (LSM780; Carl Zeiss) equipped with a 63× objective lens and ZEN software (v2009; Carl Zeiss). To avoid temporary binding or overlapping of DsRed- and YFP-labeled mitochondria, time-series images were recorded for 150 s (100 frames). The DsRed/YFP-bound mitochondria for more than 15 s (10 frames) were considered stably tethered mitochondria. The number of DsRed-labeled mitochondria stably tethered with YFP-labeled mitochondria out of total DsRed mitochondria was counted and expressed as the percentage of mitochondria tethering.

## Statistical analysis

GraphPad Prism 8.0 was used for statistical analysis and graph generation. Quantitative data were displayed as mean ± standard deviation (SD) of at least three independent experiments. Data distribution was assumed to be normal but this was not formally tested. Data were analyzed with one-way ANOVA or Student's $t$ test (two-tailed, unpaired). The P value $<0.05$ was considered statistically significant, indicated as *, $P < 0.05$; **, $P < 0.01$; and ***, $P < 0.001$.

## Online supplemental material

Fig. S1 shows NME3 acts downstream of PLD6 for mitochondrial clustering. Fig. S2 shows the sequence alignment of NME family members. Fig. S3 shows the effects of optoPLDs on mitochondrial morphology. Fig. S4 shows PLD6 is dispensable for the steady mitochondrial targeting of full-length NME3. Fig. S5 shows mitochondrial tethering assay and mitochondrial contact in swelled cells. Table S1 lists the plasmids used in this study. Table S2 lists the antibodies used in this study. Video 1 shows mitochondrial targeting of Opto-PLD induces mitochondrial clustering. Video 2 shows the enrichment of NME3 at the mitochondrial contact interface. Video 3 shows rapamycin-induced targeting of NME3$^{\Delta N}$ constructs leads to mitochondrial clustering. Video 4 shows a time-lapse image of in vitro mitochondria tethering assay with or without ectopic NME3 overexpression.

## Data availability

All data are available in the main text or the supplementary materials.

## Acknowledgments

We thank the staff of the imaging core at First Core Labs, National Taiwan University for technical support as well as members of Liu and Chang laboratories for helpful comments.

This work was supported by Ministry of Science and Technology grant MOST 107-3017-F-002-002 to Y.W. Liu, MOST 110-2636-B-007-011 and 111-2636-B-007-009 to Y.C. Lin, and National Taiwan University grant NTU-CDP-112L7809 to Y.W. Liu. J.M. Baskin acknowledges support from a Beckman Young Investigator award, a Sloan Research Fellowship, and the National Science Foundation (CAREER CHE-1749919). R. Tei was supported by Funai Overseas and Cornell Fellowships. Open Access funding provided by the National University of Taiwan.

Author contributions: Conceptualization: Y.-A. Su, H.-Y. Chiu, Y.-C. Chang, Y.-W. Liu, and Z.-F. Chang; Methodology: Y.-A. Su, H.-Y. Chiu, Y.-C. Chang, Y.-C. Lin, R. Tei, H. Bauer, Y. Feng, and J.M. Baskin; Investigation: Y.-A. Su, H.-Y. Chiu, Y.-C. Chang, C.-W. Chen, S.-S. Lin, X.-R. Huang, H.-C. Wang, C.-W. Chen, and J.-C. Hsu; Supervision: Y.-C. Lin, J.M. Baskin, Y.-W. Liu, and Z.-F. Chang; Writing—Y.-C. Chang, H.-Y. Chiu, and Y.-A. Su; Writing—review & editing: Y.-A. Su, Y.-W. Liu, and Z.-F. Chang.

Disclosures: All authors have completed and submitted the ICMJE Form for Disclosure of Potential Conflicts of Interest.

J. Baskin reported a patent to WO2023107929A1, Engineered phospholipase D mutants, methods of making engineered phospholipase D mutants, and uses thereof pending. No other disclosures were reported.

Submitted: 20 January 2023

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

**Supplemental material**

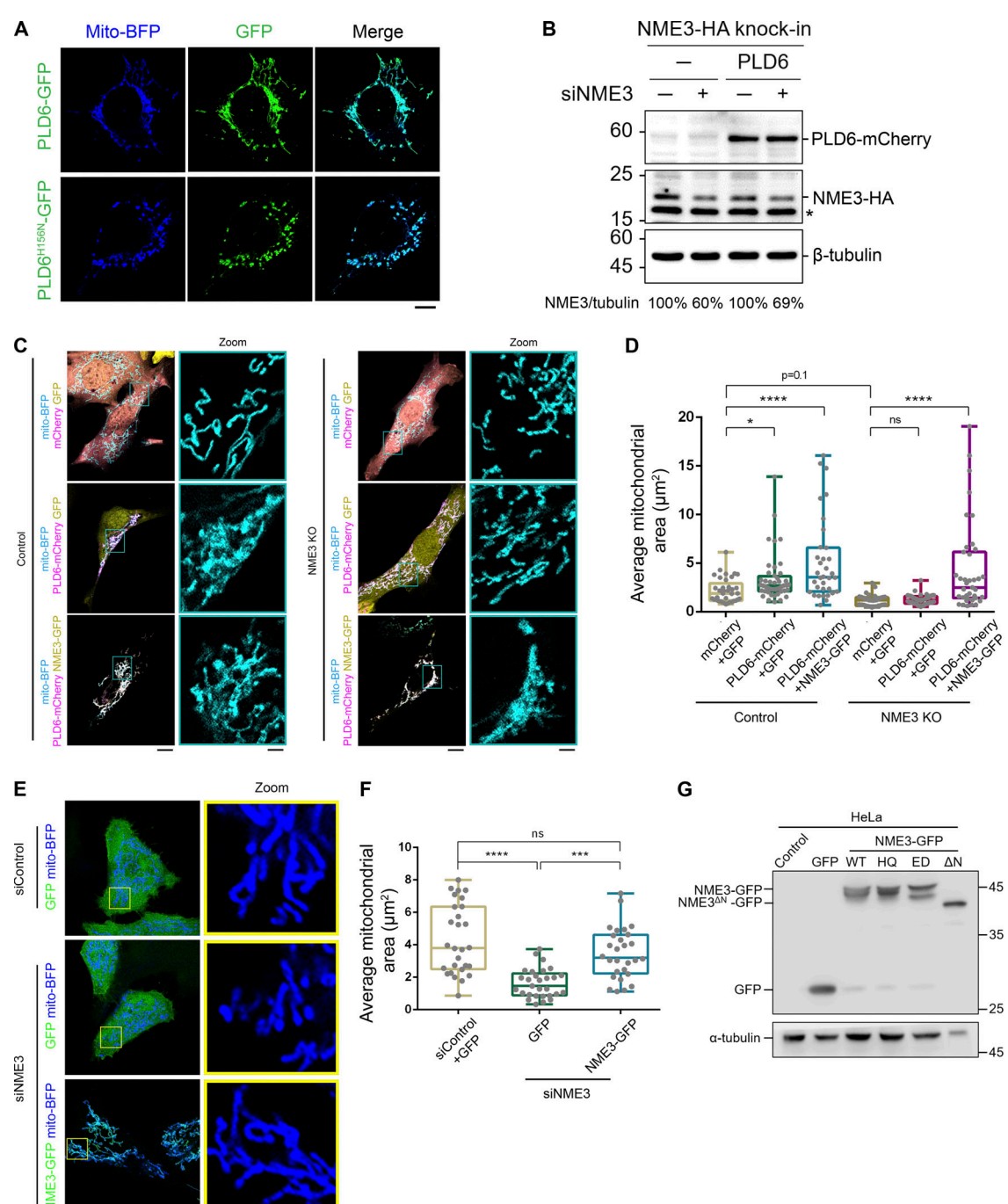

Figure S1. **NME3 acts downstream of PLD6 for mitochondrial clustering. (A)** Effects of PLD6-EGFP overexpression on mitochondria. Wild-type or catalytic dead (H156N) PLD6-EGFP were co-transfected with mito-BFP into HeLa cells. While PLD6-EGFP expression results in mitochondrial clustering, the expression of PLD6^H156N-EGFP lead to mitochondrial fragmentation. Scale bar, 10 μm. **(B)** Knockdown efficiency of siNME3. Due to the lack of a specific antibody for NME3, we generated HA knock-in NME3 HeLa cell to monitor the effect of NME3 depletion by siRNA SMARTPool (Dharmacon). The knock down efficiency was measured by Western blotting with anti-HA antibody, indicated by arrow, and normalized with tubulin. Star indicates a non-specific signal. Comparable expression of PLD6-mCherry was observed in control and NME3 knockdown cells. **(C)** NME3 is required for the mitochondrial clustering effect induced by PLD6-mCherry overexpression. Control or NME3 knockout mouse embryonic fibroblasts were transfected to overexpress mCherry, EGFP, PLD6-mCherry, or NME3-GFP. Scale bar, 10 μm. Bar for inset images, 2 μm. **(D)** The mitochondria in these cells were labeled by mito-BFP, and the average area of individual mitochondrium was measured by ImageJ and shown in D. Each dot represents the averaged mitochondrium area of one cell. **(E and F)** Rescuing ability of NME3-GFP in siNME3 cells. Wild-type NME3-GFP was transfected into siNME3 HeLa cells. The rescuing abilities of these constructs were demonstrated by the significant restoration of the average size of mitochondria shown in F. Scale bar, 10 μm. Bar for inset images, 2 μm. **(G)** The expression level of different NME3-GFP constructs. HeLa cells transfected with indicated NME3-GFP constructed were blotted with anti-GFP antibody. Due to its high expression level, less cell lysate of NME3^ΔN-GFP transfected cell was loaded in doer to avoid saturated signal of GFP. Data are expressed as mean ± SD of three independent experiments and analyzed with one-way ANOVA. *P < 0.05; **P < 0.01; ***P < 0.001; ****P < 0.0001; ns, not significant. Source data are available for this figure: SourceData FS1.

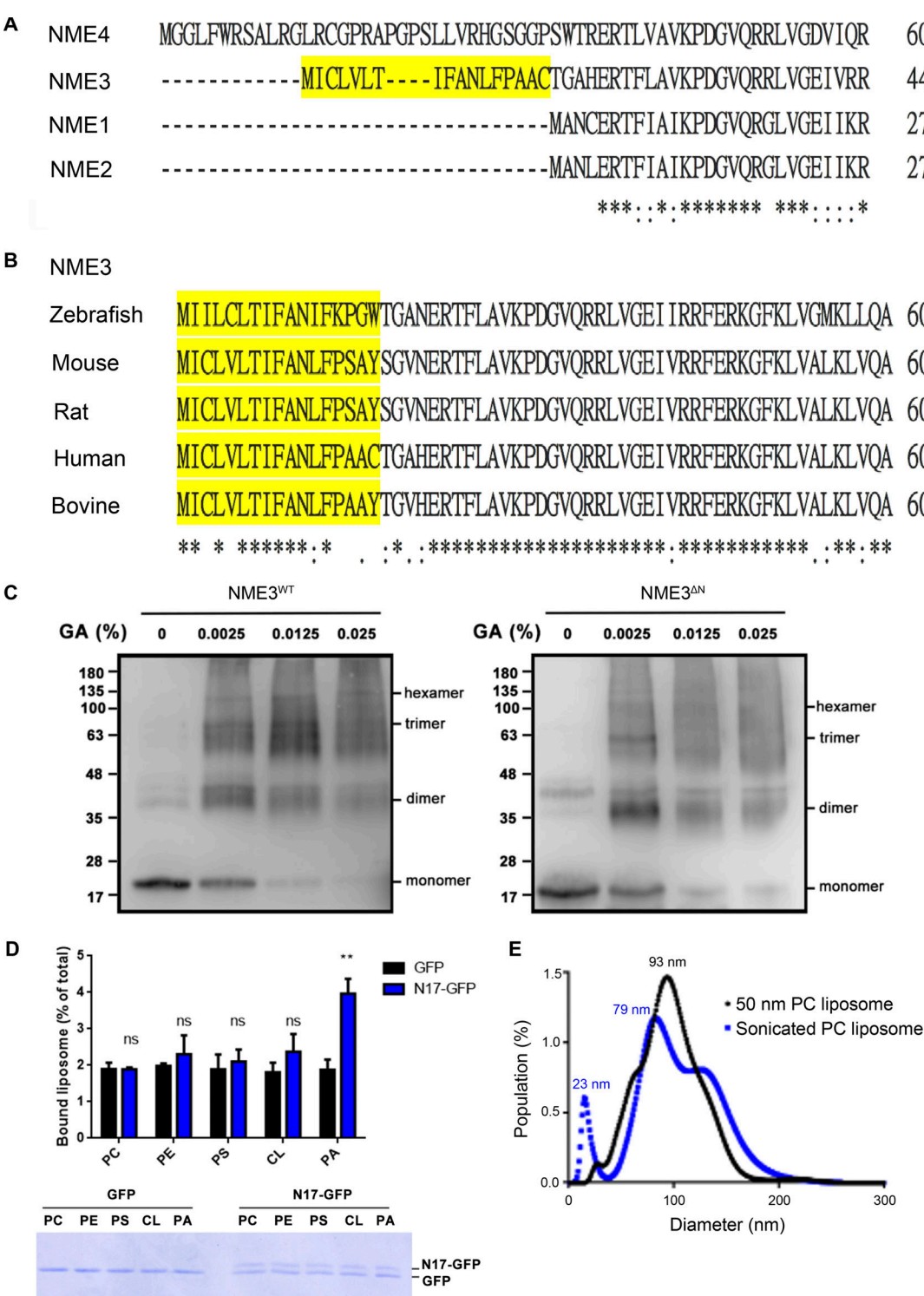

Figure S2. **Sequence alignment NME family members. (A)** Amino acid sequence alignment of the N-terminal region of human NME family members. The N17 amino acid region of NME3 is marked in yellow. These sequences were analyzed with EMBL-EBI Clustal Omega Multiple Sequence Alignment. **(B)** Sequence alignment of the N-terminal region of NME3 from different organisms. **(C)** Glutaraldehyde crosslinking experiment. To analyze the oligomerization status of recombinant NME3$^{WT}$-His and NME3$^{\Delta N}$-His, 4 µM purified proteins were subjected to crosslinking using glutaraldehyde, forcing the formation of covalent bonds, and detected by Western blotting. **(D)** Liposome binding ability of N17-GFP. To examine the direct binding between N17 and lipid, N17-GFP was purified from N17-GFP transfected HeLa cells with GFP-trap, and then subjected to liposome binding composed of 1% rhodamine-PE, 79% PC, and 20% indicated lipid. After PBS wash, bound liposomes were quantified as remained fluorescent intensity. Upper panel shows the fluorescent intensity of bound liposomes, where the lower gel shows the amount of protein in each binding reaction. N = 3. **(E)** Size distribution of PC liposomes prepared by sonication (SUV) or extrusion through membrane with 50 nm pore. Data are expressed as mean ± SD of three independent experiments and analyzed with one-way ANOVA. **P < 0.01; ns, not significant. Source data are available for this figure: SourceData FS2.

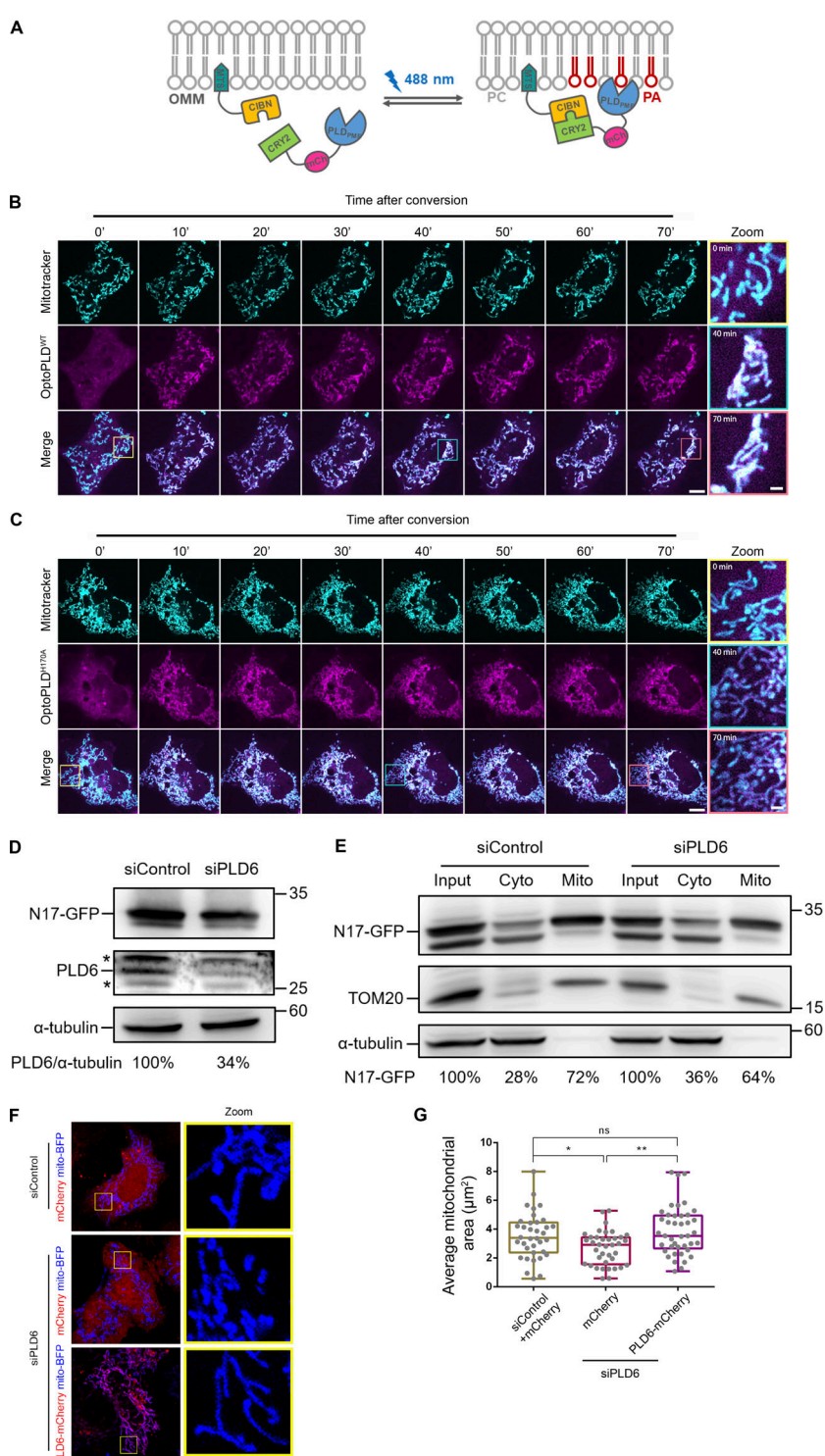

Figure S3. **Effects of optoPLDs on mitochondrial morphology. (A)** Scheme of the optoPLD assay. **(B and C)** The increase of PLD activity induces mitochondrial clustering. optoPLD$^{WT}$ or optoPLD$^{H170A}$ were transfected into Cos-7 cells, and these cells were stained with mitotracker deep red prior to conversion. The optoPLDs (magenta) expressing cell was subjected to photo-conversion for 1 s per 2.5 min during the 70-min image acquisition. Boxed areas were enlarged and shown on the right panel. Scale bar, 10 μm. Bar for inset images, 2 μm. **(D)** The expression level of N17-GFP in control and PLD6-depleted cells. **(E)** Subcellular distribution of N17-GFP in PLD6-depleted cells. HeLa cells were co-transfected with N17-GFP as well as control or PLD6 siRNA SMARTPool. After 72 h, cells were subjected to fractionation to measure the mitochondrial targeting ratio of N17-GFP. Similar to their distribution observed under fluorescent microscopy, the amount of N17-GFP in mitochondrial fraction is slightly decreased by the depletion of PLD6. **(F and G)** Rescuing ability of PLD6-mCherry in siPLD6 cells. Wild-type PLD6-mCherry was transfected into siPLD6 HeLa cells. The rescuing abilities of these constructs were demonstrated by the significant restoration of average size of mitochondria shown in G. Scale bar, 10 μm. Bar for inset images, 2 μm. Data are expressed as mean ± SD of three independent experiments and analyzed with one-way ANOVA. *P < 0.05; **P < 0.01; ns, not significant. Source data are available for this figure: SourceData FS3.

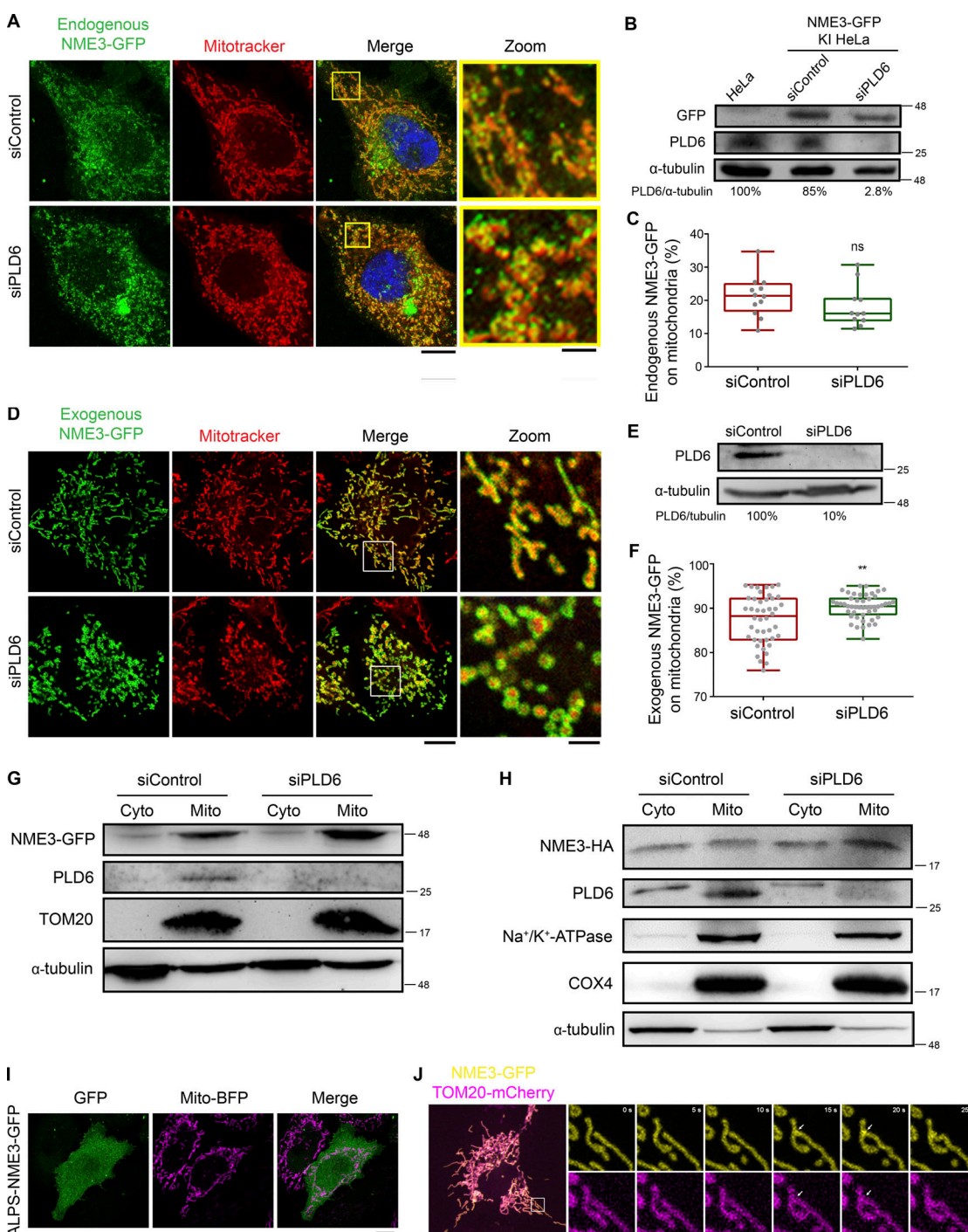

Figure S4.  **PLD6 is dispensable for the steady mitochondrial targeting of full-length NME3. (A)** Effect of PLD6 depletion on the subcellular localization of endogenous NME3. HeLa cells with GFP knock-in to the C-terminus of NME3 were transfected with PLD6 siRNA SMARTPool. **(B and C)** The knock-down and mitochondrial localization efficiency of endogenous NME3-GFP were quantified by Western blotting, confocal microscopy, and ImageJ analysis. **(D)** Effect of PLD6 depletion on the subcellular localization of exogenous NME3. NME3-GFP was transfected into control or PLD6-depleted HeLa cells. **(E and F)** The knock-down and mitochondrial localizing efficiency of ectopic NME3-GFP were quantified by Western blotting, confocal microscopy, and ImageJ analysis. **(G and H)** Subcellular distribution of NME3. HeLa cells with GFP or HA knock-in to the C-terminus of NME3 were transfected with control or PLD6 siRNA SMARTPool. After 72 h, cells were subjected to fractionation to measure the mitochondrial targeting ratio of endogenous NME3 with two different tags. Similar to their distribution observed under fluorescent microscopy, the amount of NME3-GFP or NME3-HA in mitochondrial fraction was not decreased by the depletion of PLD6. **(I)** Localization of ALPS-NME3-EGFP. The chimeric NME3-EGFP with its N17 replaced by the amphipathic helix lipid packing sensor (ALPS) derived from the PA-binding domain of Spo20 (62–79 amino acid) was expressed in HeLa cells and imaged together with mito-BFP. **(J)** The enrichment of NME3-GFP at mitochondrial contact interface prior to fusion. The dynamic distribution of NME3-GFP in two contact mitochondria was observed in Cos-7 cells co-transfected with NME3-GFP and Tom20-mCherry, which labels mitochondrial outer membrane. Arrows indicate the contact interface between two closely positioned mitochondria. Scale bars, 10 μm. Bars for inset images, 2 μm. Data are expressed as mean ± SD of three independent experiments and analyzed with Student's *t* test. **P < 0.01; ns, not significant. Source data are available for this figure: SourceData FS4.

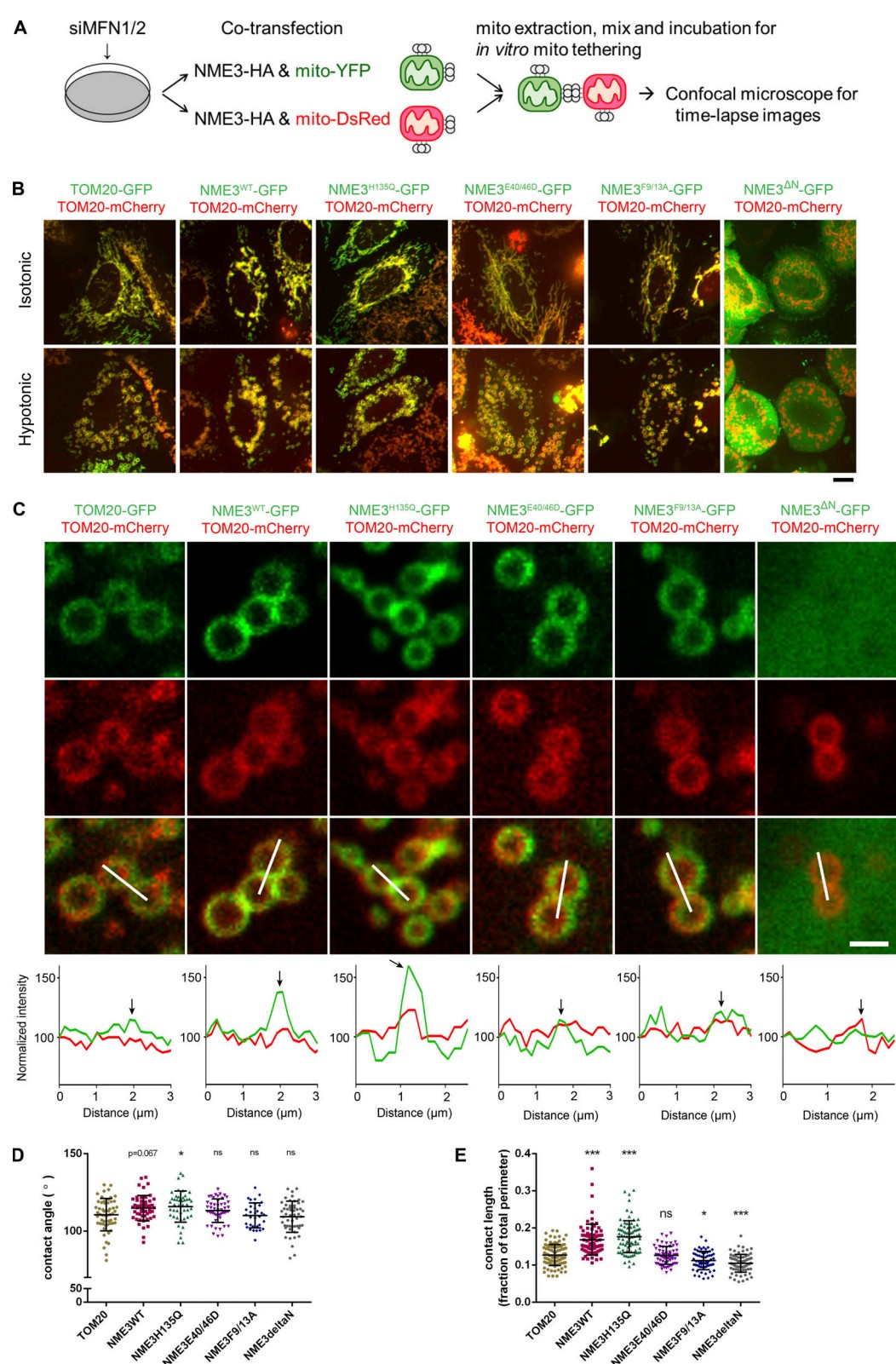

Figure S5. **Mitochondrial tethering assay and mitochondrial contact in swelled cells. (A)** Scheme of the in vitro mitochondrial tethering assay. HEK293T cells were transfected with siRNA against MFN1/2 (siMFN1/2). After 2 d, cells were co-transfected with an equal amount of NME3-HA and mito-YFP (or mito-DsRed). Isolated mitochondria labeled with YFP and DsRed were mixed for in vitro tethering. **(B)** Mitochondria large intracellular vesicles (LICVs) distribution in exogenous NME3 expressing cells. Hela cells expressing indicated Tom20-GFP or NME3-GFP together with Tom20-mCherry were incubated in 5% DMEM for 10 min. Cells before and after the hypotonic swelling were imaged and shown. Scale bar, 10 μm. **(C)** The representative mitochondrial LICVs were magnified and shown in C, scale bar of 2 μm. **(D and E)** The contact angel and length between closely positioned mitochondrial LICVs were quantified by ImageJ and shown in D and E, respectively. Data are expressed as mean ± SD and analyzed with one-way ANOVA. *P < 0.05; ***P < 0.001; ns, not significant.

Video 1.   **Mitochondrial targeting of Opto-PLD induces mitochondrial clustering.** optoPLD$^{WT}$ or optoPLD$^{H170A}$ were transfected into Cos-7 cells, and these cells were stained with mitotracker deep red prior to conversion. Time-lapse images were acquired before and after blue-light-induced mitochondrial targeting. Cells were continuously shined with blue light by pulse with 90-s intervals.

Video 2.   **Enrichment of NME3 at mitochondrial contact interface.** Cos-7 cell transfected with NME3-GFP was imaged under spinning disc confocal microcopy with a 5-s interval. The boxed area was magnified.

Video 3.   **Rapamycin-induced targeting of NME3$^{\Delta N}$ constructs leads to mitochondrial clustering.** Cos-7 cells co-transfected with Tom20-CFP-FRB and indicated Neon-FKBP-ΔNNME3 constructs were imaged upon 100 nM rapamycin addition with confocal microscopy.

Video 4.   **Time-lapse image of in vitro mitochondria tethering assay with or without ectopic NME3 overexpression.** Isolated DsRed- or YFP-labeled mitochondria with or without exogenous NME3$^{WT}$ or NME3$^{E40/46D}$ expression are mixed together and then imaged with confocal microscopy to track their motion.

**Provided online are Table S1 and Table S2. Table S1 lists the plasmids used in this study. Table S2 lists the antibodies used in this study.**

