## [Peer Review File · The Journal of Cell Biology]

NME3 binds to phosphatidic acid and mediates PLD6-induced mitochondrial tethering

You-An Su, Hsin-Yi Chiu, Yu-Chen Chang, Chieh-Ju Sung, Chih-Wei Chen, Reika Tei, Xuan-Rong Huang, Shao-Chun Hsu, Shan-Shan Lin, Hsien-Chu Wang, Yu-Chun Lin, Jui-Cheng Hsu, Hermann Bauer, Yuxi Feng, Jeremy Baskin, Zee-Fen Chang, and Ya-Wen Liu

Corresponding Author(s): Ya-Wen Liu, National Taiwan University

Review Timeline:

Submission Date:	2023-01-20
Editorial Decision:	2023-03-01
Revision Received:	2023-06-10
Editorial Decision:	2023-07-13
Revision Received:	2023-07-14

Monitoring Editor: Laura Lackner

Scientific Editor: Tim Fessenden

Transaction Report:

DOI: <https://doi.org/10.1083/jcb.202301091>

March 1, 2023

Re: JCB manuscript #202301091

Prof. Ya-Wen Liu
National Taiwan University
Institute of Molecular Medicine
No. 1, Sec. 1, Jen-Ai Rd., R1517
Taipei 10002
Taiwan

Dear Prof. Liu,

Thank you for submitting your manuscript entitled "NME3 binds to phosphatidic acid and mediates PLD6-induced mitochondrial tethering". The manuscript has been evaluated by expert reviewers, whose reports are appended below. Unfortunately, after an assessment of the reviewer feedback, our editorial decision is against publication in JCB at this time.

You will see that reviewers appreciated the conceptual advance on mitochondrial fusion through NME3, supported with high quality experiments and imaging. However, multiple important claims made in this work were felt to be insufficiently supported by the data shown. Namely, conclusions made Figure 3 should be bolstered with quantification as noted by multiple reviewers, and the question on Figure 1 raised by Reviewer 1 should be addressed. Examining other lipids (point B raised by Reviewer 3) is important to clarify specificity of these findings. This Reviewer also sought clarification on the impact of N17 mutants on NME3 hexamerization (point G). Finally, reporting the expression levels of mutants and overexpressed proteins was requested by Reviewer 2 and we agree this would be important. While a revision would should address all reviewer comments must in some form, additional data beyond those noted here are not required. We agree with Reviewer 3 that replacing N17 with a similar domain (they suggest ALPS) would make an intriguing confirmation of these observations and if possible we encourage the authors to include these data.

Given interest in the topic, I would be open to resubmission to JCB of a significantly revised and extended manuscript that fully addresses the reviewers' concerns noted above and is subject to further peer-review. If you would like to resubmit this work to JCB, please contact the journal office to discuss an appeal of this decision or you may submit an appeal directly through our manuscript submission system. An appeal must also include a point-by-point plan for revision on which we may consult with reviewers. Please note that priority and novelty would be reassessed at resubmission.

Regardless of how you choose to proceed, we hope that the comments below will prove constructive as your work progresses. We would be happy to discuss the reviewer comments further once you've had a chance to consider the points raised in this letter. You can contact the journal office with any questions, cellbio@rockefeller.edu or call (212) 327-8588.

Thank you for thinking of JCB as an appropriate place to publish your work.

Sincerely,

Laura Lackner
Monitoring Editor
Journal of Cell Biology

Tim Fessenden
Scientific Editor
Journal of Cell Biology

Reviewer #1 (Comments to the Authors (Required)):

This manuscript identifying NNE3 as binding to PA and mediating PLD6-induced mitochondrial tethering is really quite nice and convincing for the most part. The authors use multiple methods to provide evidence for their findings, show very nice and clear data, provide a well-organized and written manuscript, and use mostly appropriate methods for analysis, quantitation, and interpretation of the data in a rigorous and reproducible manner. I have several small and one substantial comment, but overall, this is a very nice story.

1) Abstract: Line 29: It is an over-statement to say that PLD6 is "essential" for mitochondrial fusion. Please see Huang et al, 2011, PMID: PMC3061402. PLD6^{-/-} mice are viable, which is not the case for mice lacking Mfn, and while there are effects on mitochondrial morphology indicating a role for PLD6/PA, the phenotype is not as strong as for MFN^{-/-} cells and mice. It would be better to say that PLD6/PA facilitate fusion.

2) Line 52; should add Choi et al (2006) and probably Huang et al 2011 here. Also, line 51 - "important" rather than "crucial" would be better.

3) Line 60 - "essential", see point (1).

4) Fig. 3F,G; line 142-3. I'm a little concerned about selection bias here. The authors highlight a representative area for zooming in the siControl cell but then highlight a "worst case" area in the siPLD6 cell. I'm not sure how different the numbers would have been if they were based on a selection from the top of the cell, which has much less cytoplasmic N17-GFP. Can the authors clarify their methods for obtaining the individual data points in Fig. 3G? Was the entire cell analyzed, or just a selected region as shown in 3F? Were the cells and regions selected by someone blinded to the experimental conditions? This is important for assessment of robustness of the data.

5. Fig. 3C-E; line 140-142. Similarly, in this experiment, all of the N17-GFP is already localized on the mitochondria at time 0 in the PLD-H170A expressing cell, whereas that is not the case in the PLD-wt expressing cell, where there's substantial N17-GFP in the cytoplasm. Hence the change in recruitment to mitochondria after PLD activation and PA generation may reflect whether there was anything left to recruit, rather than whether PA was generated. Perhaps these are not representative cells, but some clarification of this issue would be good.

6. Line 140. Probably "in parallel" or "consistently" would be better here than "Coincidentally".

7. General comment. This report would be stronger if CRISPR/Cas were used to knockout PLD6 in the cell lines for the key experiments, rather than using siRNA which is only knocking down PLD6 by 75%. I think the findings are adequate as is; but would strongly recommend to the reviewers that they generate and use this resource for future studies.

Reviewer #2 (Comments to the Authors (Required)):

NME3 is a nucleoside diphosphate kinase that produces nucleoside triphosphates such as GTP by transferring phosphate from ATP. NME3 has been proposed to function in mitochondrial fusion. However, there is relatively limited available data supporting its role in fusion compared to dynamin-related GTPases, such as Mfn1/2 and Opa1. In this current work, the authors analyzed the function of NME3 in mitochondrial aggregation induced by the overexpression of PLD6, a mitochondrial phospholipase that produces phosphatidic acid and regulates mitochondrial fusion. However, it is unclear whether the mitochondrial aggregation induced by PLD6 overexpression is relevant to mitochondrial fusion since this aggregation is independent of the enzymatic activity of PLD6. Therefore, the significance of this work is limited. Also, some experiments lack the necessary controls and robust readouts, as described below.

Specific comments

In Figure 1, the authors overexpressed PLD6 and induced mitochondrial aggregation and then tested if NME3 knockdown affects the aggregation. In contrast to previous reports, PLD6 overexpression only aggregated a small fraction of mitochondria in cells, and the current experimental setup is not robust. Therefore, although the authors claim that NME3 knockdown reduced aggregation, the data are not convincing. Was PLD6 sufficiently overexpressed? The authors should test the levels of PLD6 overexpression in the presence or absence of NME3 knockdown. For NME3 knockdown, a control with re-expression of NME3 is necessary to rule out off-target effects.

In the same figure, the authors tested different NME3 mutants on mitochondrial aggregation. The authors conclude that WT NME3 aggregated mitochondria, while its E40D and E46D mutants did not. However, this could be due to the different expression levels of the mutants. The authors should confirm the expression levels of WT and mutant constructs.

In Figure 2, the authors show that the first 17 amino acids (NME N17) bind to PA in vitro and are localized to mitochondria in cells. These are fascinating data.

In Figure 3, the authors claim that the optogenetic recruitment of a bacterial WT PLD, but not a catalytic dead mutant, induces mitochondrial aggregation. However, in contrast to the authors' claim, clear aggregation of mitochondria is not seen.

Based on the imaging in Figures 3F and G, the authors claim that the mitochondrial localization of NME N17 is decreased in PLD6 knockdown cells. Many control experiments are missing. For example, are the expression levels of NME N17 similar in control and knockdown cells? Can PLD6 re-expression rescue the knockdown effect? The authors should biochemically test the localization of NME N17-GFP.

In Figures 4 and 6, the authors suggest that NME-GFP is enriched in mitochondria-mitochondria contact sites. But this enrichment is difficult to see in the presented images. To rule out the possibility that this is not due to increased amounts of the outer membrane, the authors should compare Tom20-BFP (or RFP) (Tom-BFP and NEM3-GFP are separately expressed in the current manuscript) and NME3-GFP in the same mitochondria, not a matrix-targeted BFP or Mitotracker, which are not outer membrane makers.

The authors describe that this NME-GFP enrichment at the contact sites depends on PLD6 based on its knockdown in Figure 4. A rescue control with PLD6-reexpression should be included to exclude off-target effects.

Reviewer #3 (Comments to the Authors (Required)):

This is a highly interesting study, which invokes a role for NME3 in stress-induced mitochondrial tethering via its putative, purportedly specific, interactions with the nonbilayer-prone, fusogenic lipid, phosphatidic acid (PA) produced by PLD6 at the mitochondrial surface. The cell biology experiments are very well executed and the imaging is absolutely top-notch. However, there are critical issues pertaining to the major conclusion of this study, as outlined below that need a thorough redress before further consideration.

As realized by the authors in the Discussion, highly curved membrane tubes and associated lipid packing defects (including those created by PA in its vicinity) are prevalent throughout the cell (e.g. in endocytic pit necks, recycling endosome tubes). This cannot explain why NME3 is targeted to mitochondrial contact sites, specifically via the hydrophobic N17 segment, in vivo. This is a major conclusion of this study, which at this juncture remains unconvincing.

- a. Can the authors replace the N17 segment with a biochemically similar, but unrelated motif, such as another ALPS motif and confirm mitochondrial specificity?
- b. Even though PA and CL are both nonbilayer-prone, cone-shaped lipids, PA has a different intrinsic curvature than CL. The pKa of (de)protonation varies as well. In other words, the charge densities on the headgroups can be different based on the environmental conditions. Other negatively charged lipids like PS, PG, PI4P and PI4,5P2 should be included in the lipid screen to ascertain PA specificity. How does the presence of PE, a prevalent nonbilayer-prone, but zwitterionic, lipid at the mitochondria, influence lipid specificity?
- c. NME3 is believed to associate with Drp1, the fission enzyme, which localizes to ER-mito contact sites and interacts directly with PLD6 (Adachi et al, 2016). Drp1, in addition to CL, also selectively binds PA. Could Drp1 be responsible for NME localization indirectly? This could explain why PLD6 depletion also reduces N17 binding to mitochondria. Would the removal of Drp1 (via knockout) influence the purported activities of NME and PLD6 at contact sites? This should be fully addressed.
- d. Could CCCP/FCCP treatment, known to measurably promote CL externalization, augment NME3-mediated mitochondrial tethering in the absence of Drp1? Would PLD6 activity alone suffice under these conditions? These experiments will support the nutrient starvation experiments, which unfortunately has many more pleiotropic effects.
- e. What does the N17 polypeptide pull down in a pull-down or a cross-linking (BioID) experiment? Could it be a specific protein binding partner rather than PA in situ? This cannot be left to speculation in the Discussion as this is the major point of this study.
- f. Extruded liposome diameters from 50 nm polycarbonate membranes and SUVs should also be confirmed by DLS. Inclusion of negatively curved, cone-shaped lipids tends to make the smaller liposomes not conform to expected diameters. Negatively curvature-preferring lipids counteract high positive curvature. Does NME also tether larger liposomes with PA?
- g. Sticky proteins, such as the NME hexamer with a hydrophobic N-terminus, can potentially cluster liposomes non-specifically. Again, appropriate controls (replacing the N17 with a biochemically similar, but unrelated, motif, and using liposomes of a different target lipid composition) is needed. Does the hexamer-defective mutant NME3E40/46D mutant bind membranes with the same affinity as WT NME? How stable is the deltaN17 mutant in folding and hexamerization? Otherwise, these are not appropriate controls.
- h. A direct measure of membrane interaction/insertion may be required for the N17 segment.

Point-by-point response to reviewers:

JCB manuscript #202301091

"NME3 binds to phosphatidic acid and mediates PLD6-induced mitochondrial tethering"

Reviewer #1 (Comments to the Authors (Required)):

This manuscript identifying NNE3 as binding to PA and mediating PLD6-induced mitochondrial tethering is really quite nice and convincing for the most part. The authors use multiple methods to provide evidence for their findings, show very nice and clear data, provide a well-organized and written manuscript, and use mostly appropriate methods for analysis, quantitation, and interpretation of the data in a rigorous and reproducible manner. I have several small and one substantial comment, but overall, this is a very nice story.

>> We deeply appreciate the recognition and constructive suggestions from the reviewer. We are particularly grateful for reviewer's careful reading and precise comments on wording. The detailed responses for each comments are described below.

1) Abstract: Line 29: It is an over-statement to say that PLD6 is "essential" for mitochondrial fusion. Please see Huang et al, 2011, PMCID: PMC3061402. PLD6^{-/-} mice are viable, which is not the case for mice lacking Mfn, and while there are effects on mitochondrial morphology indicating a role for PLD6/PA, the phenotype is not as strong as for MFN^{-/-} cells and mice. It would be better to say that PLD6/PA facilitate fusion.

>> We thank the reviewer for the correction. We have edited the sentence as in Line 28, "Phosphatidic acid (PA) on mitochondrial outer membrane generated by PLD6 facilitates the fusion of mitochondria".

2) Line 52; should add Choi et al (2006) and probably Huang et al 2011 here. Also, line 51 - "important" rather than "crucial" would be better.

>> We agree with the reviewer and have edited the sentence accordingly (Line 49, "While these GTPases are the driving machinery for membrane remodeling, two non-bilayer-forming phospholipids, cardiolipin (CL) and phosphatidic acid (PA), have been found to be important for the fission and fusion processes of mitochondria (Choi et al., 2006; Huang et al., 2011; Kameoka et al., 2018; Osman et al., 2011).").

3) Line 60 - "essential", see point (1).

>> We have edited this word. Thank you very much. (Line 58, "Intriguingly, CL can

be hydrolyzed and converted into PA by mitoPLD/PLD6, a phospholipase D on the outer membrane of mitochondria, where PA functions as a signaling lipid for mitochondrial clustering and facilitates fusion”)

4) Fig. 3F,G; line 142-3. I'm a little concerned about selection bias here. The authors highlight a representative area for zooming in the siControl cell but then highlight a "worst case" area in the siPLD6 cell. I'm not sure how different the numbers would have been if they were based on a selection from the top of the cell, which has much less cytoplasmic N17-GFP. Can the authors clarify their methods for obtaining the individual data points in Fig. 3G? Was the entire cell analyzed, or just a selected region as shown in 3F? Were the cells and regions selected by someone blinded to the experimental conditions? This is important for assessment of robustness of the data.

>> We apologize for the lack of clear description of mitochondrial localizing efficiency of N17-GFP in legend of Figure 3, although it was briefly mentioned in Methods. We added the description in the legend for Fig. 3 (Line 588, “To quantify the mitochondrial targeting efficiency, the total intensity of N17-GFP on mitochondria in a cell was quantified by ImageJ, which was divided by the total N17-GFP intensity of a cell. Each dot represents the ratio of one cell.). Thus, the answer is Yes, the entire cell was analyzed for the mitochondrial targeting efficiency of the proteins of interest. Since the image capturing and analysis were not conducted by someone blinded to the experiments, we further include a biochemical approach, the subcellular fractionation, to support the partial decrease of mitochondrial targeting of N17-GFP in PLD6 depleted cells (Sup Fig S3E).

5. Fig. 3C-E; line 140-142. Similarly, in this experiment, all of the N17-GFP is already localized on the mitochondria at time 0 in the PLD-H170A expressing cell, whereas that is not the case in the PLD-wt expressing cell, where there's substantial N17-GFP in the cytoplasm. Hence the change in recruitment to mitochondria after PLD activation and PA generation may reflect whether there was anything left to recruit, rather than whether PA was generated. Perhaps these are not representative cells, but some clarification of this issue would be good.

>> We deeply appreciate the helpful suggestion. We have replaced the figure with a better representative cells that shows comparably low level of N17-GFP at mitochondria at time 0 in PLD-wt and PLD-H170A expressing cell (Fig 3C, D).

6. Line 140. Probably "in parallel" or "consistently" would be better here than "Coincidentally".

>> We agree with the reviewer and have edited the sentence (Line 149, “In parallel, we observed a significant increase in N17-GFP targeting to mitochondria upon activation of opto-PLD^{WT}, but not the catalytic dead opto-PLD^{H170A}”). Many thanks.

7. General comment. This report would be stronger if CRISPR/Cas were used to knockout PLD6 in the cell lines for the key experiments, rather than using siRNA which is only knocking down PLD6 by 75%. I think the findings are adequate as is; but would strongly recommend to the reviewers that they generate and use this resource for future studies.

>> Once again, we thank the reviewer for the constructive and helpful suggestion. We will definitely use PLD6 KO cells for our future study.

Reviewer #2 (Comments to the Authors (Required)):

NME3 is a nucleoside diphosphate kinase that produces nucleoside triphosphates such as GTP by transferring phosphate from ATP. NME3 has been proposed to function in mitochondrial fusion. However, there is relatively limited available data supporting its role in fusion compared to dynamin-related GTPases, such as Mfn1/2 and Opa1. In this current work, the authors analyzed the function of NME3 in mitochondrial aggregation induced by the overexpression of PLD6, a mitochondrial phospholipase that produces phosphatidic acid and regulates mitochondrial fusion. However, it is unclear whether the mitochondrial aggregation induced by PLD6 overexpression is relevant to mitochondrial fusion since this aggregation is independent of the enzymatic activity of PLD6. Therefore, the significance of this work is limited. Also, some experiments lack the necessary controls and robust readouts, as described below.

>> We thank the reviewer for the constructive comments. We have carefully addressed the specific comments and responded in detailed below. Besides, regarding the effect of PLD6 overexpression, our results show the catalytic activity of PLD6 is indeed required for its effect on mitochondrial cluster phenotype (Supplementary Figure S1A), suggesting this PLD6-induced mitochondrial clustering is relevant to their fusion. Together with other new results described below, we hope the reviewer will find the revised manuscript compelling to support our conclusion.

Specific comments

In Figure 1, the authors overexpressed PLD6 and induced mitochondrial aggregation and then tested if NME3 knockdown affects the aggregation. In contrast to previous reports, PLD6 overexpression only aggregated a small fraction of mitochondria in cells, and the

current experimental setup is not robust. Therefore, although the authors claim that NME3 knockdown reduced aggregation, the data are not convincing. Was PLD6 sufficiently overexpressed? The authors should test the levels of PLD6 overexpression in the presence or absence of NME3 knockdown. For NME3 knockdown, a control with re-expression of NME3 is necessary to rule out off-target effects.

>> In this revision, Sup Fig S1B showed that NME3 knockdown did not affect the level of PLD6 overexpression. In Sup Fig S1E and F, we showed the re-expression of NME3-GFP in siNME3 transfected cells restored mitochondrial morphology. Furthermore, similar results were observed in NME3 KO MEFs overexpressing PLD6-mCherry, which could be reversed by NME3-GFP re-expression (Sup Fig S1C,D).

In the same figure, the authors tested different NME3 mutants on mitochondrial aggregation. The authors conclude that WT NME3 aggregated mitochondria, while its E40D and E46D mutants did not. However, this could be due to the different expression levels of the mutants. The authors should confirm the expression levels of WT and mutant constructs.

>> We thank the reviewer for the constructive comments. The expression levels of NME3 mutants and WT by Western blotting is shown in Supplementary figure S1G, which indicates that similar levels of proteins were expressed.

In Figure 2, the authors show that the first 17 amino acids (NME N17) bind to PA in vitro and are localized to mitochondria in cells. These are fascinating data.

>> We thank the reviewer for the warm and positive comment. We are extremely grateful.

In Figure 3, the authors claim that the optogenetic recruitment of a bacterial WT PLD, but not a catalytic dead mutant, induces mitochondrial aggregation. However, in contrast to the authors' claim, clear aggregation of mitochondria is not seen.

>> We agree with the reviewer that the mitochondrial phenotype after opto-PLD conversion is relatively mild, and is partially clustering. We thus have softened our description into “we found that mitochondria became more clustered after 70 min of photo-activation of opto-PLD^{WT}, whereas a catalytic-dead opto-PLD^{H170A} did not induce significant clustering” (Line 147).

Based on the imaging in Figures 3F and G, the authors claim that the mitochondrial localization of NME N17 is decreased in PLD6 knockdown cells. Many control experiments

are missing. For example, are the expression levels of NME N17 similar in control and knockdown cells? Can PLD6 re-expression rescue the knockdown effect? The authors should biochemically test the localization of NME N17-GFP.

>> We thank the reviewer for the constructive comments. We have added a WB result to show comparable N17-GFP expression level in control and PLD6 knockdown cells (Sup Figure S3D), together with the data that rule out siRNA off-target by re-expression of PLD6 (Sup Fig S3F-G). We also added the cell fractionation data to show the localization of N17-GFP (Sup Figure S3E). All these results support the partial contribution of PLD6 on N17-GFP targeting to mitochondria.

In Figures 4 and 6, the authors suggest that NME-GFP is enriched in mitochondria-mitochondria contact sites. But this enrichment is difficult to see in the presented images. To rule out the possibility that this is not due to increased amounts of the outer membrane, the authors should compare Tom20-BFP (or RFP) (Tom-BFP and NEM3-GFP are separately expressed in the current manuscript) and NME3-GFP in the same mitochondria, not a matrix-targeted BFP or Mitotracker, which are not outer membrane makers.

>> We thank the reviewer for the constructive comments. The image analysis of Tom20-mCherry and NME3-GFP in the same mitochondria showed that only the intensity of NME3, but not Tom20, was increased at the mitochondrial contact sites. We have added this result in Sup Figure S4J.

The authors describe that this NME-GFP enrichment at the contract sites depends on PLD6 based on its knockdown in Figure 4. A rescue control with PLD6-reexpression should be included to exclude off-target effects.

>> We thank the reviewer for the constructive comments. We have added the rescue control with PLD6 re-expression result to exclude off-target effects in Sup Figure S3F-G.

Reviewer #3 (Comments to the Authors (Required)):

This is a highly interesting study, which invokes a role for NME3 in stress-induced mitochondrial tethering via its putative, purportedly specific, interactions with the nonbilayer-prone, fusogenic lipid, phosphatidic acid (PA) produced by PLD6 at the mitochondrial surface. The cell biology experiments are very well executed and the imaging is absolutely top-notch. However, there are critical issues pertaining to the major conclusion of this study, as outlined below that need a thorough redress before further consideration.

As realized by the authors in the Discussion, highly curved membrane tubes and associated

lipid packing defects (including those created by PA in its vicinity) are prevalent throughout the cell (e.g. in endocytic pit necks, recycling endosome tubes). This cannot explain why NME3 is targeted to mitochondrial contact sites, specifically via the hydrophobic N17 segment, in vivo. This is a major conclusion of this study, which at this juncture remains unconvincing.

>> We thank the reviewer for these constructive comments. We have carefully addressed these issues as described below. We hope the reviewer will find the revised manuscript clear and strong enough to support our conclusion.

a. Can the authors replace the N17 segment with a biochemically similar, but unrelated motif, such as another ALPS motif and confirm mitochondrial specificity?

>> We thank the reviewer for this interesting idea. As suggested, we have replaced an ALPS motif from the PA binding domain of Spo20 (residue 62-79, Horchani et al., 2014, PLOS ONE) and observed a mainly cytosolic distribution (Sup Figure S4I). This result demonstrates the mitochondrial specificity of N17 of NME3, and indicates that PA binding is insufficient for mitochondrial localization. This result has been added to Sup Fig S4I and discussed in Discussion session (Line 281-291).

b. Even though PA and CL are both nonbilayer-prone, cone-shaped lipids, PA has a different intrinsic curvature than CL. The pKa of (de)protonation varies as well. In other words, the charge densities on the headgroups can be different based on the environmental conditions. Other negatively charged lipids like PS, PG, PI4P and PI4,5P2 should be included in the lipid screen to ascertain PA specificity. How does the presence of PE, a prevalent nonbilayer-prone, but zwitterionic, lipid at the mitochondria, influence lipid specificity?

>> We agree with the reviewer thus have added PS and PE into the N17-GFP-lipid binding analysis to ascertain PA specificity. This result has been included in Sup Fig S2D.

c. NME3 is believed to associate with Drp1, the fission enzyme, which localizes to ER-mito contact sites and interacts directly with PLD6 (Adachi et al, 2016). Drp1, in addition to CL, also selectively binds PA. Could Drp1 be responsible for NME localization indirectly? This could explain why PLD6 depletion also reduces N17 binding to mitochondria. Would the removal of Drp1 (via knockout) influence the purported activities of NME and PLD6 at contact sites? This should be fully addressed.

>> We have expressed NME3 in Drp1 KO MEFs and found NME3 perfectly localized on mitochondria, indicating Drp1 is not essential for NME3 mitochondrial binding (Figure below). Since mitochondrial network is highly connected without dynamics and with some aggregation in Drp1 KO cells, it would be difficult to assess NME3 enrichment at the interface contact sites using time-lapse recording. Our in vitro liposome assays indicated that NME3 by itself is capable of PA binding. Since NME3 depletion markedly increases mitochondrial fragmentation, it is more tentatively to assume the function of NME3 is linked to PLD6-regulated fusion. We totally agree that NME3 can interact with Drp1. Whether the spatial regulation of NME3 by PLD6 is involved in the stress-induced fission by Drp1 should be interesting for further investigation.

Drp1 is not essential for the mitochondria localization and clustering ability of NME3 and PLD6.

PLD6-mCherry and NME3-GFP were transfected together with mito-BFP into control or Drp1 KO MEFs. Confocal micrographs were shown. Scale, 10 μm.

d. Could CCCP/FCCP treatment, known to measurably promote CL externalization, augment NME3-mediated mitochondrial tethering in the absence of Drp1? Would PLD6 activity alone suffice under these conditions? These experiments will support the nutrient starvation experiments, which unfortunately has many more pleiotropic effects.

>> We thank the reviewer for the constructive comments. Consistent with the reviewer's speculation, we observed augmented NME3 enrichment at mitochondrial interface upon CCCP or UV treatment. We are currently investigating this phenomenon in more details to understand the physiological meaning. Given that PLD6-mCherry and NME3-EGFP over-expression could still result in mitochondrial clustering in Drp1 KO fibroblast (Figure above), it is apparent that the mitochondrial tethering function of NME3 and PLD6 is independent of Drp1.

e. What does the N17 polypeptide pull down in a pull-down or a cross-linking (BioID) experiment? Could it be a specific protein binding partner rather than PA in situ? This cannot be left to speculation in the Discussion as this is the major point of this study.

>> We agree with the reviewer that PA binding is not enough for N17-GFP localizing to mitochondria. The replacement of N17 with the PA-binding ALPS derived from Spo20 results in cytosolic distribution of chimeric NME3 (Sup Fig S4I). Furthermore, we found that N17-GFP could associate with Drp1 in a pull down analysis (data not shown). These results demonstrate that specific protein binding is more likely responsible for the recruitment of N17 to mitochondria, and PA binding mainly contributes to the enrichment of NME3 at the interface of mitochondria. This notion is carefully described in Discussion (Line 281-291).

f. Extruded liposome diameters from 50 nm polycarbonate membranes and SUVs should also be confirmed by DLS. Inclusion of negatively curved, cone-shaped lipids tends to make the smaller liposomes not conform to expected diameters. Negatively curvature-preferring lipids counteract high positive curvature. Does NME also tether larger liposomes with PA?

>> We thank the reviewer for the constructive comments. The liposomes prepared from extrusion through 50 nm polycarbonate membrane or sonication are composed of 100 % DOPC (Fig 2F). The DLS confirmation data has been included into the Sup Figure S2E. As for NME3 tethering ability, we indeed utilized liposomes extruded with 1,000 nm polycarbonate membrane for microscopy analysis in Figure 5D, which show relatively low tethering ability than 100-nm liposome shown in Figure 5E.

g. Sticky proteins, such as the NME hexamer with a hydrophobic N-terminus, can potentially cluster liposomes non-specifically. Again, appropriate controls (replacing the N17 with a biochemically similar, but unrelated, motif, and using liposomes of a different target lipid composition) is needed. Does the hexamer-defective mutant NME3E40/46D mutant bind

membranes with the same affinity as WT NME? How stable is the deltaN17 mutant in folding and hexamerization? Otherwise, these are not appropriate controls.

>> Thanks for the constructive comments. To further examine the specificity of NME3 toward PA, we have included PS and PE into the lipid binding analysis (Sup Fig S2D). In addition, the hexamer-defective mutant NME3^{E40/46D} mutant binds PA liposome with comparable affinity as WT (Fig 2C). Together, these results show the PA-specific binding and tethering ability of NME3.

On the other hand, the deltaN17 mutant is stable (based on our expression and purification experiences) and hexamerized, shown by the cross-linking experiment in Sup Fig S2C.

h. A direct measure of membrane interaction/insertion may be required for the N17 segment.

>> We thank the reviewer for the constructive comments. Given the difficulty to express and purify N17 from bacteria, we purify N17-GFP from HeLa cells, and conduct liposome binding analysis with N17-GFP immobilized on GFP-trap (Sup Fig S2D). These results show that N17-GFP binds to PA liposome, but not PC, PE, PS nor CL liposomes.

July 13, 2023

RE: JCB Manuscript #202301091R-A

Prof. Ya-Wen Liu
National Taiwan University
Institute of Molecular Medicine
No. 1, Sec. 1, Jen-Ai Rd., R1517
Taipei 10002
Taiwan

Dear Prof. Liu:

Thank you for submitting your revised manuscript entitled "NME3 binds to phosphatidic acid and mediates PLD6-induced mitochondrial tethering". We would be happy to publish your paper in JCB pending final revisions necessary to meet our formatting guidelines (see details below). Please also attend to the request from Reviewer 2 as you undertake these final revisions.

A. MANUSCRIPT ORGANIZATION AND FORMATTING:

Full guidelines are available on our Instructions for Authors page, <http://jcb.rupress.org/submission-guidelines#revised>. Submission of a paper that does not conform to JCB guidelines will delay the acceptance of your manuscript.

1) Text limits: Character count for Articles is < 40,000, not including spaces. Count includes abstract, introduction, results, discussion, and acknowledgments. Count does not include title page, figure legends, materials and methods, references, tables, or supplemental legends.

2) Figures limits: Articles may have up to 10 main figures and 5 supplemental figures/tables.

3) Figure formatting: Scale bars must be present on all microscopy images, including inset magnifications. Molecular weight or nucleic acid size markers must be included on all gel electrophoresis. Please avoid pairing red and green for images and graphs to ensure legibility for color-blind readers. If red and green are paired for images, please ensure that the particular red and green hues used in micrographs are distinctive with any of the colorblind types. If not, please modify colors accordingly or provide separate images of the individual channels.

** Please include scale bars for inset images throughout, as well as for heatmaps in Figure 3D.

** Please add molecular weight markers to gel blots in Figure 2, Figure 4E,

4) Statistical analysis: Error bars on graphic representations of numerical data must be clearly described in the figure legend. The number of independent data points (n) represented in a graph must be indicated in the legend. Statistical methods should be explained in full in the materials and methods. For figures presenting pooled data the statistical measure should be defined in the figure legends. Please also be sure to indicate the statistical tests used in each of your experiments (either in the figure legend itself or in a separate methods section) as well as the parameters of the test (for example, if you ran a t-test, please indicate if it was one- or two-sided, etc.). Also, if you used parametric tests, please indicate if the data distribution was tested for normality (and if so, how). If not, you must state something to the effect that "Data distribution was assumed to be normal but this was not formally tested."

** Please include error bar descriptions in the caption for each figure, and note statistical tests used.

5) Abstract and title: The abstract should be no longer than 160 words and should communicate the significance of the paper for a general audience. The title should be less than 100 characters including spaces. Make the title concise but accessible to a general readership.

6) Materials and methods: Should be comprehensive and not simply reference a previous publication for details on how an experiment was performed. Please provide full descriptions in the text for readers who may not have access to referenced manuscripts. We also provide a report from SciScore and an associate score, which we encourage you to use as a means of evaluating and improving the methods section.

** Please include a description of the optoPLD construct (such as an AddGene reference) rather than referencing a prior publication.

** Please ensure sufficient details on transmission EM methods are provided.

7) Please be sure to provide the sequences for all of your primers/oligos and RNAi constructs in the materials and methods. You must also indicate in the methods the source, species, and catalog numbers (where appropriate) for all of your antibodies. Please also indicate the acquisition and quantification methods for immunoblotting/western blots.

8) Microscope image acquisition: The following information must be provided about the acquisition and processing of images:

- a. Make and model of microscope
- b. Type, magnification, and numerical aperture of the objective lenses
- c. Temperature
- d. Imaging medium
- e. Fluorochromes
- f. Camera make and model
- g. Acquisition software
- h. Any software used for image processing subsequent to data acquisition. Please include details and types of operations involved (e.g., type of deconvolution, 3D reconstitutions, surface or volume rendering, gamma adjustments, etc.).

10) Supplemental materials: There are strict limits on the allowable amount of supplemental data. Articles may have up to 5 supplemental figures. Please also note that tables, like figures, should be provided as individual, editable files. A summary of all supplemental material should appear at the end of the Materials and methods section.

13) ORCID IDs: ORCID IDs are unique identifiers allowing researchers to create a record of their various scholarly contributions in a single place. At resubmission of your final files, please consider providing an ORCID ID for as many contributing authors as possible.

Please note that JCB now requires authors to submit Source Data used to generate figures containing gels and Western blots with all revised manuscripts. This Source Data consists of fully uncropped and unprocessed images for each gel/blot displayed in the main and supplemental figures. Since your paper includes cropped gel and/or blot images, please be sure to provide one Source Data file for each figure that contains gels and/or blots along with your revised manuscript files. File names for Source Data figures should be alphanumeric without any spaces or special characters (i.e., SourceDataF#, where F# refers to the associated main figure number or SourceDataFS# for those associated with Supplementary figures). The lanes of the gels/blots should be labeled as they are in the associated figure, the place where cropping was applied should be marked (with a box), and molecular weight/size standards should be labeled wherever possible. Source Data files will be directly linked to specific figures in the published article.

Journal of Cell Biology now requires a data availability statement for all research article submissions. These statements will be published in the article directly above the Acknowledgments. The statement should address all data underlying the research presented in the manuscript. Please visit the JCB instructions for authors for guidelines and examples of statements at (<https://rupress.org/jcb/pages/editorial-policies#data-availability-statement>).

WHEN APPROPRIATE: The source code for all custom computational methods published in JCB must be made freely available as supplemental material hosted at www.jcb.org. Please contact the JCB Editorial Office to find out how to submit your custom macros, code for custom algorithms, etc. Generally, these are provided as raw code in a .txt file or as other file types in a .zip file. Please also include a one-sentence summary of each file in the Online Supplemental Material paragraph of your manuscript.

B. FINAL FILES:

Thank you for this interesting contribution, we look forward to publishing your paper in Journal of Cell Biology.

Sincerely,

Laura Lackner
Monitoring Editor
Journal of Cell Biology

Tim Fessenden
Scientific Editor
Journal of Cell Biology

Reviewer #1 (Comments to the Authors (Required)):

The authors were responsive to my comments.

Reviewer #2 (Comments to the Authors (Required)):

The authors have successfully addressed all of my comments by conducting critical control experiments. I fully support the publication of this compelling and exciting work. As a minor point, I suggest the authors provide information regarding silent mutations in the rescue constructs used in the siRNA knockdown experiments for NME3 and PLD6.

Reviewer #3 (Comments to the Authors (Required)):

The authors have satisfactorily addressed my concerns with additional data or text clarifications. I have no further issues.